



# Modelling the effects of climate and landcover change on the hydrologic regime of a snowmelt-dominated montane catchment

Russell S. Smith[1], Caren C. Dymond[2], David L. Spittlehouse[3], Rita D. Winkler[3], and Georg Jost[4]

[1]WaterSmith Research Inc., Kelowna, V1Y 5N3, Canada
[2]Ministry of Forests, Victoria, V8W 9C5, Canada
[3]Retired, Ministry of Forests, Victoria, V8W 9C5, Canada
[4]Generation System Operations, BC Hydro, Burnaby, V3N 4X8, Canada

*Correspondence to*: Russell S. Smith (rsmith@watersmith.ca)

**Abstract.** Climate change poses risks to society through the potential to alter peak flows, low flows, and annual runoff yield. Wildfires are projected to increase due to climate change; however, little is known about their combined effects on hydrology. This study models the combined impacts of climate and landcover changes on the hydrologic regime of a snowmelt-dominated montane catchment, to identify management strategies that mitigate negative impacts. The combination of climate change and stand replacing landcover disturbance in the middle and high elevations is predicted to advance the timing of the peak flow two to nine times (depending on emission pathway) more than the advance generated by disturbance alone. The modelling predicts that the combined impacts of climate change and landcover disturbance on peak flow magnitude are generally offsetting for events with return periods less than 5-25 years, but additive for more extreme events. There is a dependency of extreme peak flows on the distribution of landcover. The modelling predicts an increasing importance of rainfall in controlling peak flow response under a changing climate, at the expense of snowmelt influence. Extreme summer low flows are predicted to become commonplace in the future, with most of the change in frequency occurring by the 2050s. Low annual yield is predicted to become more prevalent by the 2050s, but then fully recover or become less prevalent (compared to the current climate) by the 2080s, because of increased precipitation in the fall-spring period. The modelling suggests that landcover disturbance can have a mitigative influence on low water supply. The mitigative influence is predicted to be sustained under a changing climate for annual water yield, but not for summer low flow. The study results demonstrate the importance of examining complexity in three dimensions with respect to modelling changes to the hydrological regime: climate change, landcover change, and numerous hydrological indicators. Moreover, for managing watershed risk, the results indicate there is a need to carefully evaluate the interplay among environmental variables, the landscape, and the values at risk. Strategies to reduce one risk may increase others, or effective strategies may become less worthwhile in the future.

Keywords: Raven Hydrological Modelling Framework, risk, hazard, land use, land-use change





# 1        Introduction

Climate change poses risks to society due to potential increases in flooding and reduced water supply (Burn et al., 2016; Gaur et al., 2019). Concurrently, wildfires are projected to increase in severity and extent in many parts of the world due to climate
change, with potentially devastating hydrologic and geomorphic consequences (Flannigan et al., 2009; Wang et al., 2015). Few studies have addressed the combined impacts of climate and landcover changes on hydrologic response, with even fewer also addressing multiple aspects of the runoff regime, including peak flows, low flows, and overall runoff yield. Doing so would facilitate a holistic evaluation of potential long-term changes in hydrology and threats to communities, infrastructure, and aquatic habitat, as well as the potential to identify adaptation strategies.

Climate change in snowmelt-dominated mountainous regions is likely to result in decreased peak snowpack depths, earlier spring snowmelt, more frequent winter rain events, and more persistent midwinter melt, resulting in more transient snowpacks and higher rates of winter runoff (Elsner et al., 2010; Merritt et al., 2006; Vano et al., 2010). Shallower snowpacks and earlier melt often result in lower peak flows, a shift towards less frequent flooding, and decreased summer low flows and overall yield
(Burn et al., 2010; Dierauer et al., 2018; Merritt et al., 2006). These studies illuminate potential hydrologic impacts related to large scale (spatial and temporal) changes in air temperature and precipitation; however, few studies have accounted for potential increases in the intensity of weather patterns at synoptic or event scales. Westra et al. (2013) showed that nearly two-thirds of rainfall stations globally exhibited increasing trends in annual maximum daily precipitation between 1900 and 2009, with corresponding increases in air temperature. The higher latitudes showed stronger positive associations, with values
ranging between approximately 7.5% and 13% °K$^{-1}$ in the Northern Hemisphere above 50ºN. Donat et al. (2016) had similar findings, with climate projections for the rest of the century showing continued intensification, particularly for dry regions. We suggest that more intense rainfall during periods of high catchment wetness (e.g., ripe snowpacks and/or wet soils) could increase flood frequencies and overall runoff yield.

Flooding and water supply are also affected by landcover change resulting from wildfire, as well as insects, disease, and forest harvesting (Robinne et al., 2021; Saxe et al., 2019; Schnorbus and Alila, 2013). Forest cover disturbance typically increases snowpack accumulation and ablation rates, advances the timing of melt, and increases runoff yield (Winkler et al., 2017; Winkler et al., 2015). Catchments with more than 20% of the forest cover disturbed have shown peak flow increases; however, the magnitude of response is highly variable, illuminating nonlinear response behaviour (Adams et al., 2012; Goeking and
Tarboton, 2020; Schnorbus and Alila, 2013). For instance, several studies have shown that the difference in peak flows between disturbed and undisturbed catchments decreases with increasing event magnitude (Bathurst et al., 2011a; Bathurst et al., 2011b; Moore and Wondzell, 2005), whereas some have shown the difference to increase with increasing event magnitude (Moore and Wondzell, 2005; Schnorbus and Alila, 2013). Yet others have shown a decrease in yield caused by disturbance (Adams et al., 2012; Goeking and Tarboton, 2020). Decreasing yield has typically been associated with non-stand replacing disturbance

and/or arid climates. Low flows have often been shown to increase immediately post-disturbance, followed by longer-term decreases as evapotranspiration (ET) increases with forest regeneration, though the changes are often not significant (Coble et al., 2020; Goeking and Tarboton, 2020). These findings indicate that the impacts of landcover disturbance on runoff yield are complex, with variability in the direction and magnitude of change dependent on the specific landcover and hydrometeorological conditions considered.


With respect to the combined influences of climate and landcover, a global study of large watersheds by Li et al. (2017) showed that climate and landcover played equal roles in annual yield variations. Among 67 watersheds, 51 exhibited additive effects of climate and landcover change, and 16 showed offsetting effects, with the former generating a higher risk of extreme outcomes (floods or droughts). They also found that smaller and dryer watersheds are hydrologically more sensitive to

landcover change than larger and wetter watersheds. Among forest-dominated regions of the world, Wei et al. (2018) found that the global mean variation in annual runoff due to landcover change was 30.7%, whereas 69.3% was attributed to climate change. For British Columbia, Canada, large scale mountain pine beetle infestation and salvage logging accounted for 39.0% of the variation in annual runoff, compared to 61.0% for climate change.

Given their unpredictable nature, the purpose of the current study was to assess the combined impacts of climate and landcover changes on the hydrologic regime of the greater Penticton Creek Watershed in southwest Canada, and to identify management strategies that mitigate negative impacts from either change. The catchment is located in a semi-arid, mountainous region. Vegetation ranges from grassland in the lower elevations, to dense coniferous forest at higher elevations. Wildfires are frequent (BC Government, 2019) and fire weather conditions are expected to become more severe as the climate warms (Nitschke and

Innes, 2008; Spittlehouse and Dymond, 2022). Climate envelope modelling indicates a potential for reduced canopy density in the lower elevations (Wang et al., 2012). Both flood and water supply risks are a concern, and climate and landcover changes are concerns for impacting these risks by increasing hazards (i.e., the probability of a harmful event). Meanwhile, Penticton Creek provides water to a town of 34,000 people that has a bustling tourism industry, and irrigation to wineries and fruit orchards.


To achieve the study objectives, we developed a catchment runoff model using the Raven Hydrological Modelling Framework (Raven) (Craig et al., 2020; Tsuruta and Schnorbus, 2021); combined different climate conditions with different landcover conditions in the model to assess impacts on hydrology; and evaluated implications to flooding and water supply limitations, upland water storage, and managing watershed risk (in relation to changing hazard). In phase 1, we combined five different

emission pathways with two different landcover conditions (pre-disturbance and severe disturbance) to examine overall impacts on the spring freshet hydrograph. An emission pathway that generated intermediary impacts in phase 1 was chosen for further modelling. In phase 2, we combined the selected emission pathway with five different landcover conditions for a more detailed investigation of impacts on water input dynamics, snowpack dynamics, runoff timing, and the frequency of peak



flows, low flows, and annual runoff yield (i.e., annual discharge). Each landcover condition was treated as static (i.e., no
change over time) and, thus, represents a specific example of landcover disturbance.

## 2        Methods

### 2.1        Study catchment

The study catchment for hydrologic modelling comprised the area draining to the Penticton Creek below Harris Creek
streamflow gauging station (Water Survey of Canada [WSC] Stn. 08NM170, 148 km$^2$, https://wateroffice.ec.gc.ca/search/
historical_e.html) (Fig. 1). Elevations in the catchment range from approximately 593 m to 2154 m, and drain from the
Okanagan Plateau, an area of relatively subdued mountainous topography. Stream discharge data were also used for three
headwater sub-catchments, including 240 (08NM240) and 241 Creeks (08NM241) (both southerly facing), and Dennis Creek
(08NM242) (westerly facing); and for Greyback Lake (controlled reservoir) (08NM169) (hereafter, all discharge values refer
to the main catchment outlet, unless specified otherwise). All discharge data were utilized at a daily time-step.


The underlying bedrock is comprised primarily of intrusive rock, with areas of metamorphic rock in the north and southwest.
The surface geology is dominated by glacial till. Soils fall into three main types, including sandy loam over bedrock at ~0.65
m, deep loam to ~1.2 m, and sandy loam over sand at ~0.6 m (Fig. 2) (https://sis.agr.gc.ca/cansis/index.html). A deeply incised
mainstem channel runs northeast-southwest, with tributary streams forming a dendritic drainage pattern.






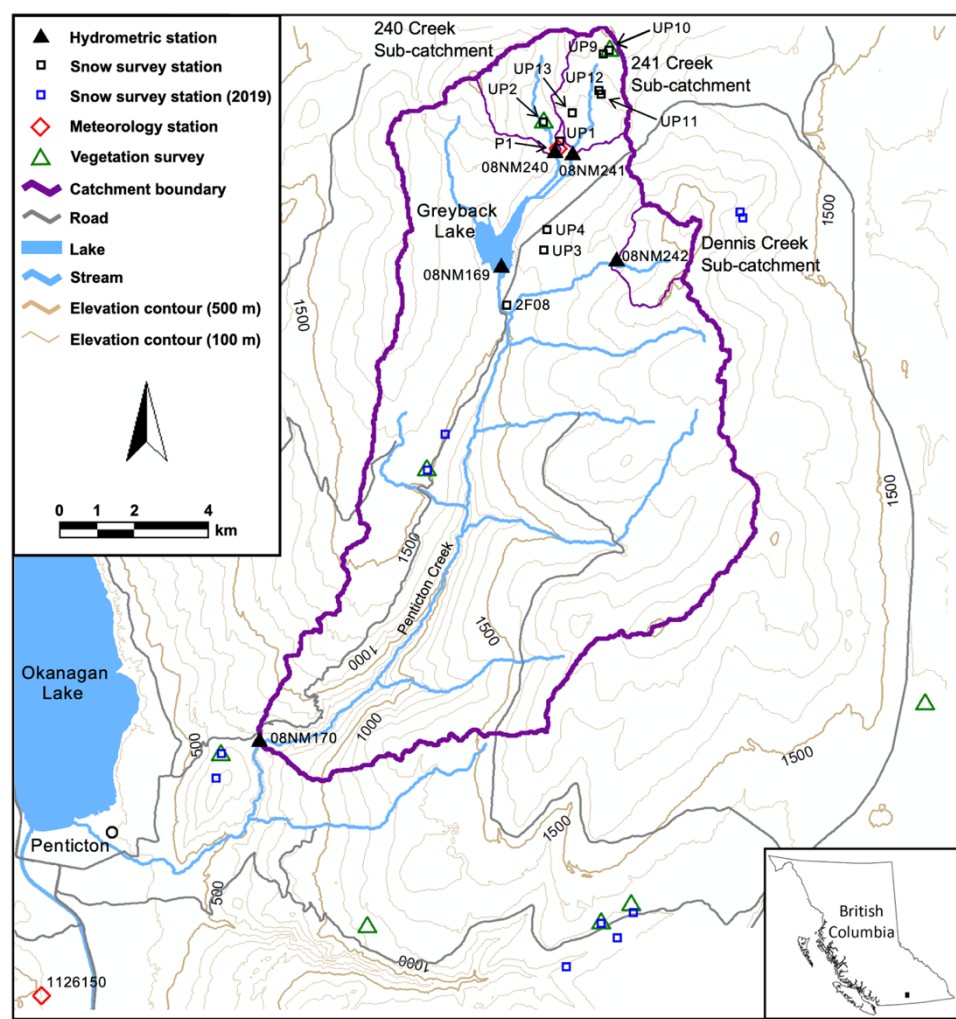

**Figure 1: Location and monitoring sites of the Penticton Creek study catchment. Snow survey station (2019) refers to data acquired by Smith (2022) (Table S2).**







**Figure 2: Physiography of the study catchment. Tree species (panel 'c') and crown closure (panel 'd') represent the estimated pre-disturbance conditions. Harvest year (panel 'f') and fire year (panel 'g') represent the actual disturbance histories.**

Five Biogeoclimatic Ecosystem Classification (BEC) zones are represented in the study catchment including Ponderosa Pine (PP), Interior Douglas-fir (IDF), Montane Spruce (MS), Engelmann Spruce – Subalpine Fir (ESSF), and Interior Mountain-heather Alpine (IMA) (Fig. 2) (Lloyd et al., 1990). The IDF and ESSF include two and three variants, respectively. Dominant tree species in the study catchment include ponderosa pine (*Pinus ponderosa*) (Py), Douglas-fir (*Pseudotsuga menziesii*) (Fd), lodgepole pine (*Pinus contorta*) (Pl), hybrid white spruce (*Picea glauca x engelmannii*) (Sxw), Engelmann spruce (*Picea engelmannii*) (Se), and subalpine fir (*Abies lasiocarpa*) (Bl).

The catchment has a history of forest cover disturbance (Fig. 2). Between 1970 and 1976, approximately 3.7% of the entire catchment was harvested, followed by 0.2% in 1984, 0.2% in 1991, and 10.4% between 1992 and 2012. Four wildfires occurred between 1919 and 1931 (covering 17% of the entire catchment), followed by fires in 1970 (30%) and 1994 (2%). The 1970

fire was located in the arid and sparsely forested lower elevations.

A snowpack persists in upper areas of the catchment from October-November through April-June, and is intermittent in lower areas. Spring snowmelt dominates the hydrologic regime, but intense rainfall events during spring freshet can also contribute substantially to peak flows. Mean annual runoff averages 265 mm.

## 2.2  Representing the catchment

### 2.2.1  Raven setup

Raven was set up as a process-oriented deterministic runoff model, and run at a daily time-step. The study catchment was spatially discretized using a quasi-distributed approach. Precipitation and air temperature were distributed from a single meteorological station, and accounted for orography and rain-snow partitioning. The snowpack balance incorporated coupled

mass and energy balance equations. The full snowpack energy balance was represented using algorithms that estimate energy fluxes using daily precipitation, and daily minimum and maximum air temperature (Quick, 1995). It accounted for cloud cover, short-wave radiation, long-wave radiation, and turbulent flux (Quick, 1995; Dingman, 2002).

Vegetation was represented as a single canopy layer, and accounted for canopy interception, canopy drip,

sublimation/evaporation of intercepted precipitation, and transpiration (Penman Monteith). In-catchment runoff was routed to the sub-catchment outlet assuming a two-layer soil. It accounted conceptually for individual controls on runoff generation, including soil evaporation, overland flow, interflow, percolation, and baseflow. Surface runoff was subsequently routed to the main catchment outlet by in-channel flow.

The historical record of streamflow at the catchment outlet was adjusted for ongoing storage changes in Greyback Lake using bathymetric and lake level data. Increasing lake storage was added to discharge for the catchment outlet (and vice versa for decreasing storage), assuming an instantaneous transfer to the catchment outlet (actual transit time during high flow is ~1 hour). This adjustment allowed the catchment to be represented as an unregulated system, thus, isolating the effects of climate and landcover change on the runoff regime without the confounding effects of reservoir operations.



### 2.2.2  Meteorology

#### 2.2.2.1  Observed record

The historical record from the P1 weather station (1619 m) in the Upper Penticton Creek Watershed Experiment (UPC) (Winkler et al., 2017) (Fig. 1) was used for interpolating air temperature and precipitation throughout the catchment using lapse rates. The P1 station has a daily record spanning 1992-2016, but was extended back to 1970 using other UPC stations (1983-1991 period) and regional records (1970-1983) (Winkler et al., 2017). The 1970-2014 portion of the extended record was used in this study to coincide with available hydrometric and snowpack records.

#### 2.2.2.2  Climate change datasets

The climate change datasets were obtained from the Bias Corrected Constructed Analogues with Quantile mapping datasets (BCCAQ-v2) (PCIC, 2014; Sobie and Murdock, 2017) for a grid cell that includes the location of the P1 weather station. These data are CMIP5 general circulation model (GCM) output (Taylor et al., 2012) of daily precipitation, and daily minimum and maximum air temperature, downscaled to 300 arc seconds (approximately 10 km). For phase 1 of the study, we selected five distinct emission pathways that bound approximately 90% of the projections for the region (Spittlehouse and Dymond, 2022), including MPI-ESM-MR-r1 RCP4.5 (MPI45), CNRM-CM5-r1 RCP4.5 (CNRM45), CSIRO-Mk3-6-0-r1 RCP8.5 (CSIRO85), MIROC-ESM-r1 RCP8.5 (MIROC85), and CanESM2-r1 RCP8.5 (CanESM85). CSIRO85 was selected for phase 2 based on the rationale provided in Section 5. Two climate conditions were utilized for each emission pathway, representing the 2041-70 period (2050s climate) and the 2071-2100 period (2080s climate). These future climate conditions were additional to a no-change baseline climate condition (current climate). Change in annual air temperature and precipitation are summarized in Table 1 for all five emission pathways.



**Table 1: Mean annual air temperature (°C) and total annual precipitation (mm) at the P1 meteorological station for the current climate, and the incremental (absolute) change projected for the five emission pathways.**

| Climate period | Emissions scenario | Air temperature (°C) | Precipitation (mm) |
|---|---|---|---|
| Current | NA | 2.0 | 785 |
| 2050s | MPI45 | +2.1 | +69 |
| | CNRM45 | +2.4 | +15 |
| | CSIR085 | +3.1 | -7 |
| | MIROC85 | +3.3 | +55 |
| | CanESM85 | +4.6 | +33 |
| 2080s | MPI45 | +2.0 | +84 |
| | CNRM45 | +2.9 | +96 |
| | CSIR085 | +5.4 | +68 |
| | MIROC85 | +5.1 | +125 |
| | CanESM85 | +6.6 | +77 |

**2.2.2.3 Synthetic record**

100 year weather records for a stationary climate (current, 2050s, and 2080s) were required for long-term hydrologic simulations and subsequent estimation of return periods for extreme events. Synthetic weather records for current and projected future conditions were created using the LARS-WG5 weather generator (Semenov and Barrow, 2013; Semenov and Stratonovitch, 2010). For the current climate, it was calibrated with 32 years (1984-2016) of daily minimum and maximum air 190 temperature and precipitation data for the P1 station. Compared to the historical observations, the simulated values tended to underestimate the statistical distribution of the coolest temperatures (Spittlehouse and Dymond, 2022). Consequently, a quantile mapping procedure was used to adjust the synthetic records (Sobie and Murdock, 2017). The procedure involved dividing the synthetic and measured temperature datasets into 5% quantiles, and regressing against each other to generate seasonally based non-linear adjustment equations. In contrast to temperature, the distribution of daily precipitation and monthly 195 1-day extreme precipitation, and the length of maximum monthly dry and wet periods were simulated well, and no quantile adjustment was required.

For the 2050s and 2080s climates, the ratio of future to base precipitation, and wet and dry period length were used in LARS-WG5 to adjust the statistical distributions of these variables. The resulting time series were assessed to ensure the mean 200 monthly change in temperature and precipitation matched that projected by the GCM. More information on the synthetic record is provided by Spittlehouse and Dymond (2022).


### 2.2.3    Landcover

The landcover conditions implemented in Raven included the historical disturbed forest cover conditions from 1976 and 2012, an estimate of the actual pre-disturbance condition (hereafter referred to as the *forested condition*), and the forested condition

combined with two simulated wildfire conditions (hereafter referred to as the *small burn* and *large burn conditions*) (Fig. 3, Table 2). The forested condition was considered the baseline condition for modelling purposes. It was reconstructed from 2012 Vegetation Resources Inventory (VRI) forest cover data (BC Government, 2012) by interpolating between mature stands after removing mapped disturbance areas. Species was gap-filled using the nearest neighbour function in SAGA GIS, and density was gap-filled using the multilevel B-spline function (Conrad et al., 2015).

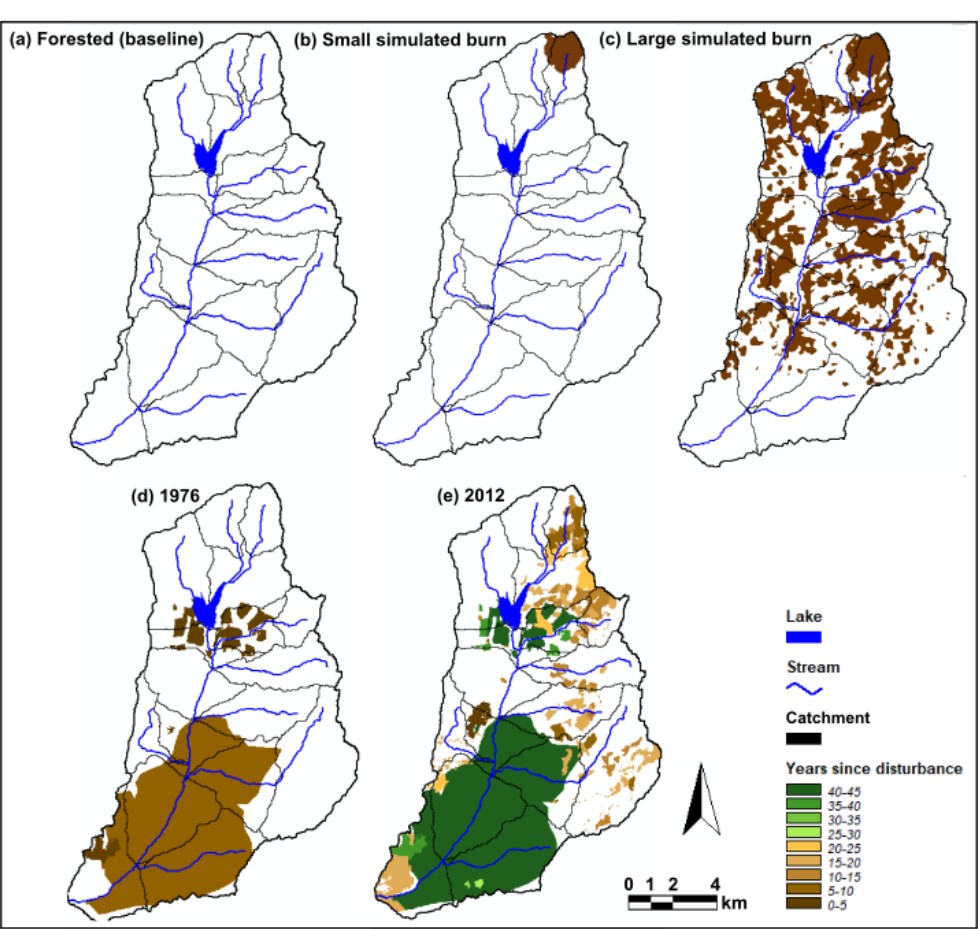

**Figure 3: Spatial representation of the historical and simulated disturbance histories implemented in Raven. Panels 'b' and 'c' represent simulated disturbances (discussed below). Panels 'd' and 'e' represent the actual disturbance histories represented in the years 1976 and 2012, respectively.**


**Table 2: Summary of disturbance area and LAI by elevation for the landcover conditions used in the long-term simulations.**

| Landcover condition | [1]Disturbed area (%) | [2]Leaf area index by elevation (m) | | | | |
|---|---|---|---|---|---|---|
| | | <1300 | 1300-1650 | 1650-1900 | >1900 | All elev. |
| Forested | [3]0 | 0.72 | 1.77 | 1.91 | 1.62 | 1.69 |
| Small burn | [4]2 | 0.72 | 1.77 | 1.86 | 1.52 | 1.66 |
| Large burn | [4]32 | 0.65 | 1.24 | 1.12 | 1.19 | 1.11 |
| 1976 | [5]34 | 0.35 | 1.21 | 1.86 | 1.62 | 1.41 |
| 2012 | [6]47 | 0.83 | 1.70 | 1.65 | 1.48 | 1.55 |

1. Portion of catchment impacted by wildfire or forest harvesting
2. HRU area weighted mean of parameter values. The percentage of watershed area comprised by each elevation range is 13, 37, 43, 7, and 100 respectively.
3. Areas burned between 1919 and 1931 were excluded because of stand regeneration.
4. Incorporates only the burned areas that were simulated as stand replacing (Sect. 2.2.3).
5. The 1970 burn occurred in areas with pre-existing stands that were naturally sparse.
6. Includes areas disturbed in the 1970s and 1980s that experienced substantial stand regeneration by 2012.

The 1976 and 2012 conditions resulted from a combination of historical wildfires and forest harvesting, with the 2012 condition incorporating several years of forest regeneration (i.e., hydrologic recovery) for older disturbance areas (Fig. 3). Some regenerating stands had higher densities than the neighbouring mature stands, resulting in the 2012 condition having higher stand densities at lower elevations than the forested condition (Table 2).

The burned areas for the small and large burn conditions were simulated using LANDIS-II, Forest Carbon Succession extension (Dymond et al., 2016), and the Dynamic Fuels Fire system (Sturtevant et al., 2009). Initial communities were based on 2014 VRI data (BC Government, 2014a). Input variables of establishment, net primary productivity, and maximum biomass were generated by Tree and Climate Assessment models (Dymond et al., 2016; Nitschke and Innes, 2008). Fire regime parameters were derived for each BEC variant (upper three variants were amalgamated due to small size) using historical fire data (points and polygons) from 1950 to 2013 over a 2.25 million ha management unit, which includes the Penticton Creek Watershed (BC Government, 2014b). A 100 m digital elevation model (DEM) was incorporated into the fire spread modelling (BC Government, 2014c). LANDIS-II was run repeatedly, taking advantage of the stochastic nature of the simulations until two scenarios occurred with a small and large burn (relative to the size of the study catchment only, as larger fires occurred in the management unit) (Fig. 3). For representing landcover patterns in Raven, only the most severe fire intensities, which caused total canopy mortality, were considered disturbed. It is acknowledged that moderate burn severities can result in substantial mortality; however, this approach was followed to limit the quantity and complexity of the landcover scenarios.





### 2.2.4 Spatial discretization

A total of 1,315 hydrological response units (HRUs) were discretized for representing the catchment (Fig. S1; Supplementary) Each of the HRUs was characterized by a unique set of parameter values representing area, elevation, vegetation, soil, slope

gradient, and slope aspect. The size of the HRUs averaged 0.11 km², with the largest HRU being 2.24 km². Table S1 (Supplementary) summarizes the spatial data used to discretize the HRUs. The spatial data (25 m grid) were processed using a combination of SAGA GIS (Conrad et al., 2015) and R statistical software (version 3.5.0) (R Core Team, 2020).

In the process of discretizing the HRUs, a total of 17 drainage units were delineated to separate the study catchment into sub-

250 catchments (Fig. 2). Additional boundaries were imprinted on the catchment discretization to distinguish BEC variant, soil type, and Greyback Lake (Fig. 2). Burned (historical and simulated) and harvested areas (Fig. 3) were also imprinted, as well as vegetation type (stratified by leading tree species and density). Finally, a 2000 m by 2000 m grid was imprinted on the discretization to limit the maximum possible size of an HRU.

For imprinting BEC variant, the IMA and ESSFdcp were combined with the slightly lower elevation ESSFdcw variant (hereafter referred to as *ESSFH*), due to the small area encompassed by the upper two variants and the lack of available data for discriminating these variants in the model parameterization. However, these upper elevation BEC variants were distinguished from the lowest elevation ESSF variant (ESSFdc1) (hereafter referred to as *ESSFL*), as the upper elevation stands have a much more clumped distribution of trees. The IDF variants and the PP were also combined (hereafter referred to as

*IDFPP*) for similar reasons as discussed above for the ESSFH. The ESSFH and IDFPP was each stratified by leading tree species and density to account for important influences on hydrology.

### 2.3 Model parameterization

For each HRU, the mean elevation, slope gradient, canopy closure, and tree height, and the median slope aspect were calculated from spatial data (Table S1). Wherever possible, other model parameters were set based on empirical observations. Data

sources included field observations and modelling experience from the study catchment and other catchments in the southern interior of BC (Smith, 2018, 2022), information available in guidance documents (Craig and the Raven Development Team, 2022; Quick, 1995), and scientific literature (examples cited below). Parameters with substantial uncertainty and/or sensitivity, and those associated with conceptually oriented algorithms (e.g., runoff routing) were set through calibrating simultaneously on in-catchment snowpack water equivalent (SWE) and stream discharge, and precipitation at Penticton Airport near the

watershed outlet (Fig. 1). Parameter ranges were constrained as permitted by available data, including the use of regional SWE data from outside the study catchment (Fig. 1; Table S2) (Smith, 2022).





The parameterization was strengthened by the use of (1) SWE data encompassing many physiographic contrasts, including elevations ranging from 710 m to 1922 m, varying hillslope orientations, clearings and mature stands, and low through high stand densities (Fig. 1; Table S2); and (2) a nested structure to the discharge data, incorporating variation in scale, catchment orientation, soil type, BEC variant, tree species, and stand density (Fig. 1). This multipronged approach provided a large amount of information for parameterizing a range of internal model processes.

### 2.3.1 Meteorology & energy balance

Several meteorological and energy balance parameters were calibrated and constrained by historical observations. The model was calibrated on long-term mean precipitation at Penticton Airport to constrain the precipitation lapse rate. Air temperature was constrained using observed lapse rates between P1 and Penticton Airport. Cloudiness was constrained based on relating solar radiation and air temperature from P1 station, and cloud penetration was constrained based on variability in the solar radiation data. Atmospheric stability was constrained based on relating air temperature and wind speed from P1 station. Albedo decay was constrained using measured snowpack albedo from UPC (Spittlehouse and Winkler, 2004). Manual snowpack observations were used to constrain snowpack patchiness during melt.

### 2.3.2 Vegetation

Some tree species were grouped for parameterizing mature stands, but were subsequently stratified by BEC variant and stand density, which resulted in the groupings at some elevations being dominated by certain species. The groupings were formed because of a tendency for stands to have a mix of species, and because of limitations in available data for parameterizing vegetation (e.g., availability of forest-clearing paired SWE for different combinations of species, density, and elevation). Groupings included Sxw, Se, and Bl as one type (spruce-fir, S/B); and Py, Fd, and Pl as another type (pine-fir, P/F). At middle and high elevations, P/F was represented primarily by Pl. At low elevations, P/F was represented primarily by Py and Fd.

The density of mature stands was classified using canopy closure values from VRI forest cover data (Table S1) as follows: clearings (<20% canopy closure), low density (20-40%) (LD), moderate density (40-68%) (MD), and high density (68-95%) (HD). Regenerating stands were classified based strictly on tree height due to a lack of stand-specific density data for the regenerating areas. Combining all factors used to differentiate stand types (species, BEC variant, canopy closure for mature stands, tree height for regenerating stands), the historical landcover conditions were represented using a total of 23 vegetation classes.

Parameter values for canopy closure and leaf area index (LAI) were informed by relating canopy closure values from the VRI forest cover data (Table S1) to canopy closure and LAI values obtained from hemispherical photos in the field. Seven plots were established in mature forests representing the following BEC variants: ESSFdc, MSdm1, IDFdm1, IDFxh1, and PPxh1. LAI is an important influence on the snowpack energy balance (and, thus, the volume and timing of snowmelt and runoff)

through extinction of solar radiation. For these reasons, LAI values were further informed through a tightly constrained
calibration on SWE and discharge. Throughfall was also calibrated on SWE (comparing forests and clearings) and discharge
for six vegetation classes that were most important for influencing the water balance. Throughfall parameters for the remaining
vegetation classes were manually assigned based on the calibrated throughfall values, accounting for differences in stand
characteristics and elevation (i.e., throughfall percentage is expected to increase with increasing precipitation). Wildfire

disturbance and clearcut harvesting were treated the same (i.e., as clearings) with respect to impacts on LAI, throughfall, and
canopy closure.

Field knowledge of local conditions and hydrometeorological processes was also considered for parameterizing vegetation.
For instance, constraints on LAI and throughfall parameters were adjusted for mature stands in the ESSFH to account for more

clumpy stand structures, compared to the ESSFL (Sect. 2.2.4). Greater wind exposure at higher elevations was also considered
for parameterizing canopy interception, due to influences on snowpack unloading from trees. Values published in literature
were also considered (Brockley and Simpson, 2004; Kollenberg and O'Hara, 1999; Pugh and Gordon, 2012).

### 2.3.3    Soils & streams

Soil mapping was used to assign soil porosity and texture, and to constrain soil depth and rates of percolation, interflow, and

baseflow (Table S1). Satellite imagery was used to assign channel widths for the Penticton Creek mainstem channel. Field
observations of the Penticton Creek mainstem channel were used to constrain channel roughness.

### 2.3.4    Calibration & validation

Various time periods were used for optimizing parameters on different data types. The 1971-1981 period was used for
optimizing on discharge at the main catchment outlet (adjusted for storage in Greyback Lake). The 1984-1992 period was used

for optimizing on sub-catchment discharge (240, 241, and Dennis Creeks). The 1995-1997 and 2009-2014 periods were used
for optimizing on SWE, to incorporate data from clearings and a range of regenerating and mature stand types. The forest
cover conditions in the catchment were relatively static over each period (Fig. 2). A split sample calibration-validation
approach was implemented. Years with high or low water yield, and cool or warm phases of the Pacific Decadal Oscillation
(PDO) were divided more-or-less evenly between calibration and validation sets.


Parameter optimization was implemented through Ostrich using the Dynamically Dimensioned Search algorithm (Matott,
2005). A composite objective function was used for evaluating model performance (i.e., goodness of fit) using the following
metrics:

- Nash-Sutcliffe Efficiency (NSE) calculated for spring freshet discharge (March-July), and again after smoothing the
data using a 15 day running mean;
- absolute bias in overall spring freshet yield;





- absolute bias in overall low flow yield (August-February);
- NSE and absolute bias for SWE; and
- absolute bias in annual precipitation at Penticton Airport.

Each metric was calculated for individual years, then averaged among years to minimize the potential for compensatory effects. Smoothed discharge data were used to emphasize the quality of the multi-day fit through the spring freshet. Absolute bias was used to ensure the model did not inflate or deflate overall yield to obtain higher NSE values. Precipitation bias was used to constrain precipitation lapse rates for achieving a plausible representation of the water balance in the lower catchment area. Spring freshet and low flow discharge were evaluated separately to ensure reasonable predictive performance for both high

and low flows. Model suitability was evaluated based on multiple factors, including the goodness of fit statistics, visual inspection of plotted output, and the plausibility of the parameter values in "physical space" (for physically oriented algorithms).

## 2.4 Long-term simulations

For phase 1 of the study, the synthetic meteorological datasets for the current climate and two future climate conditions (2050s

and 2080s) from each of the five emission pathways (Table 1) (Sect. 2.2.2) were each paired separately with the forested and large burn landcover conditions (Fig. 3; Table 2) (Sect. 2.2.3) to generate 22 different modelling scenarios. For phase 2, the current climate and the future climate conditions for CSIRO85 (Table 1) were each paired separately with the five landcover conditions (Fig. 3; Table 2) to generate 15 different modelling scenarios. Each landcover condition was treated as static during the 100 year simulation, and each year in the 100 year record represented a different variation of a given climate. Discharge

and SWE were, thus, simulated over a 100 year period for each modelling scenario. This approach allowed calculation of probability distributions from the hydrological outputs. The first year of simulation was used as a warm-up (i.e., spin-up) to ensure the soils were suitably "wet" leading into the subsequent simulation years. Accordingly, the first year of discharge output was discarded prior to analyzing.

## 2.5 Snowpack sensitivity analyses

A point-scale snowpack sensitivity analysis was implemented in phase 2 of the study to examine the influences of vegetation and topography on snowpack dynamics, and changes under the future climate conditions. Site characteristics were selected to represent topographic end-points within the range of typical conditions existing in the study catchment. The selected characteristics included south and north facing sites (i.e., high and low solar exposures) on a 40% slope gradient at elevations of 1100 m, 1500 m, and 1800 m (hereafter referred to as *low elevation*, *middle elevation*, and *high elevation*). Two vegetation

types were simulated for each elevation, including a clearing, and the most prevalent mature forest type at the specific combination of elevation and solar exposure. Results were analysed for annual maximum SWE and timing of snowpack melt-out (first day of the year with less than 25 mm of SWE).



A catchment scale snowpack sensitivity analysis was also implemented in phase 2 to further examine the influences of
vegetation, topography, and climate on snowpack dynamics. To examine the climate effect, annual maximum SWE and the
timing of snowpack melt-out were mapped catchment-wide for the forested condition under the current, 2050s, and 2080s
climates. To examine changes in the disturbance effect under a changing climate, the difference in SWE and melt-out between
100% cleared and forested catchment conditions were mapped for each climate.

## 2.6    Frequency analysis

For phase 2, intensity-duration-frequency analysis was implemented on the synthetic precipitation records, and event frequency
analysis (EFA) was implemented on output from the long-term simulations for the annual maximum discharge (i.e., peak
flow), summer low flow (lowest 30-day mean discharge between day of year 172 through 264), and annual discharge (mean
discharge over the water year). For peak flow, higher runoff was treated as more extreme in the EFA due to risks to society
(e.g., residential and commercial structures, stream crossings) and the environment (e.g., integrity of fish habitat). For annual
discharge, lower runoff (versus higher runoff) was treated as more extreme in the EFA due to risks to water supply (e.g.,
domestic consumption, ecological flows).

The intensity-duration-frequency analysis and the EFA involved statistically fitting frequency curves to annual data series
based on the Log Pearson Type III statistical distribution (using the R statistical programming language; (Craig and the Raven
Development Team, 2022). The Gringorten extreme value plotting position formula was used for plotting annual data
(Gringorten, 1963).

## 3    Model performance

The overall fit to the observed data was generally good for both the calibration and validation periods, based on visual
inspection of the plots (Fig. S2 and S3) and the performance statistics (Table S3). The composite objective function was 0.87
for the calibration and 0.79 for the validation. The overall level and timing of spring freshet runoff was matched well in most
years at the catchment and sub-catchment scales. This finding generally holds true when evaluating both early (e.g., 1971,
1988, 1992) and late (e.g., 1972, 1991) spring freshet seasons, and both the westerly oriented Dennis Creek sub-catchment and
the southerly oriented 240 Creek and 241 Creek sub-catchments. In addition, the observed snowpack data were matched well
regardless of elevation, solar exposure, or vegetation type, and regardless of the seasonal snowpack accumulation being high
(e.g., 2011-2013) or low (e.g., 2009-2010 & 2014).

Notwithstanding the good overall fit, the Dennis Creek spring freshet peaked and receded somewhat too early in 1989 and
1991 (Fig. S3.4); however, these discrepancies were small relative to the overall duration of the spring freshet, and were not
unexpected considering the simulation relied on the meteorological record that was extended using regional data. For Penticton





Creek and Dennis Creek low flows, the performance statistics decreased considerably from the calibration to the validation; however, these decreases were considered acceptable since winter hydrometric records can be subject to large errors (e.g., ice influences). Collectively, the calibration and validation results indicate good overall representation of the distribution of precipitation, air temperature, ET, snowpack processes, and fast and slow runoff response mechanisms.

## 4        Phase 1 results: all five emission pathways

Figure 4 presents spring freshet hydrographs based on the median of daily discharge from the long-term simulations, which generally represents frequently occurring discharge conditions. Comparing the five emission pathways to the current climate (i.e., climate effect) for the forested landcover condition, the simulations predicted an advance in the timing of spring freshet (both rising and falling limbs), a decrease in the peak flow discharge, and elongation of the hydrograph, for all pathways in the 2050s and most pathways in the 2080s (Fig. 4). The severity of these climate effects was generally lower for MPI45 and

CNRM45, and greater for MIROC85 and CanESM85. The severity was greater in the 2080s compared to the 2050s for most pathways. For CSIRO85, the severity was similar to MPI45 and CNRM45 in the 2050s, but shifted to being intermediary between the two groups in the 2080s. The main exceptions to these climate effects was a small increase in peak flow discharge in the 2080s compared to the current climate for CNRM45, and minimal change in the peak flow timing between the 2050s and 2080s for MPI45.


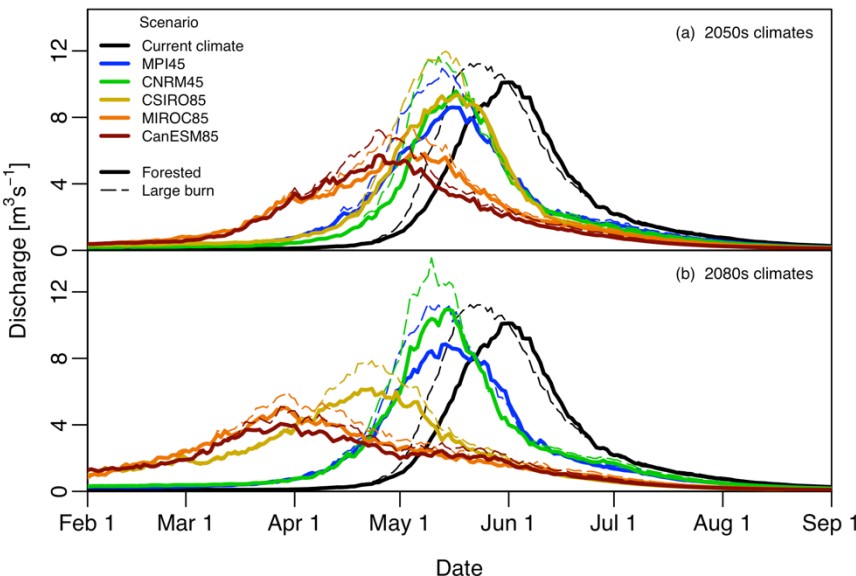

**Figure 4: Median of daily discharge for the mature forest and large burn conditions under the current climate, and under the 2050s (panel 'a') and 2080s (panel 'b') climates for the emission pathways.**

For the forested landcover condition, the mean date of the peak flow (long-term mean among all annual peak flows in the 99
year dataset) was day 151 of the year under the current climate (Fig. 4). Among the five emission pathways, the climate effect
advanced the peak flow by 15 to 37 days (overall mean of 24 days) in the 2050s, and by 15 to 60 days (overall mean of 36
days) in the 2080s. The change in the mean peak flow discharge (long-term mean among all annual peak flows in the 99 year
dataset) ranged from 0% to -30% (overall mean of -15%) in the 2050s, and from +6% to -38% (overall mean of -16%) in the
2080s. Discharge was predicted to rise substantially in the winter for MIROC85 and CanESM85 in the 2050s and 2080s, and
CSIRO85 in the 2080s.

Comparing the large burn and forested conditions for the current climate (i.e., disturbance effect), the peak flow advanced by
seven days, the mean peak flow discharge increased by 15%, and the hydrograph elongated somewhat because of the rising
limb advancing more than the falling limb (Fig. 4). Among the five emission pathways, the disturbance effect on the peak flow
timing ranged from a three day advance to a one day delay (overall mean advance of 1.8 days) in the 2050s, and from a five
day advance to a two day delay (overall mean advance of 1.2 days) in the 2080s. The mean peak flow discharge increased by
19% to 24% (overall mean of 21%) in the 2050s, and by 16% to 22% (overall mean of 19%) in the 2080s.

## 5    Phase 2 results:  CSIRO85 focus

The CSIRO85 emission pathway generated climate effects on the spring freshet hydrograph that were generally intermediary
between the other pathways for most aspects related to peak flow timing and discharge, and hydrograph elongation (based on
results in Section 4 and additional results in Section 5.2 below). For this reason, CSIRO85 was selected for the phase 2
investigation.

### 5.1    Water input dynamics

#### 5.1.1    Rainfall

Seasonal changes in air temperature and precipitation are summarized in Table 3 for CSIRO85. The pathway projects warmer
weather all year, combined with a drier summer in the 2050s and 2080s, a wetter fall in the 2050s, and a wetter fall-spring
period in the 2080s. The combined effects of the projected increase in air temperature and a shift in the seasonal distribution
of precipitation for the future climate conditions resulted in increased winter and spring rainfall (based on the rain-snow
partitioning simulated in the model), and decreased summer rainfall (Table 4), despite only small changes in overall winter
and spring precipitation for the 2050s (Table 3). The relative change in spring rainfall intensity increased with increasing event
duration. Changes in the 2080s were +18%, +24%, and +29% for 1-, 3-, and 14-day durations (based on data in Table 4).





**Table 3: Mean air temperature (°C) and total precipitation (i.e., rainfall and snowfall) (mm) by season for the current climate at the P1 and Penticton Airport meteorological stations, and the incremental (absolute) change projected for the future climates (CSIRO85) at the P1 station (adapted from Spittlehouse and Dymond, 2022; all values derived from 1983-2016 meteorological records).**

| Station (climate condition) | Air temperature (°C) | | | | Precipitation (mm) | | | |
| --- | --- | --- | --- | --- | --- | --- | --- | --- |
| | Fall | Winter | Spring | Summer | Fall | Winter | Spring | Summer |
| P1 (Current) | -1.3 | -5.3 | 3.5 | 11.2 | 205 | 186 | 240 | 161 |
| Penticton Airport (Current) | 5.6 | 0.8 | 12.1 | 19.7 | 86 | 78 | 113 | 86 |
| P1 (2050s) | +3.1 | +2.4 | +2.1 | +4.0 | +32 | +0 | +8 | -57 |
| P1 (2080s) | +4.9 | +5.1 | +4.3 | +6.7 | +67 | +31 | +17 | -63 |

**Table 4: Mean rainfall (i.e., excluding snowfall) (mm) for the current climate at the P1 meteorological station, and the incremental (absolute) change projected for the future climates (CSIRO85). Winter and summer values are the long-term mean of the seasonal total. Spring values are the long-term mean of the seasonal maximum for the specified duration.**

| Climate condition | Winter | Spring | | | Summer |
| --- | --- | --- | --- | --- | --- |
| | | 1 day | 3 days | 14 days | |
| Current | 10 | 22 | 31 | 60 | 143 |
| 2050s | +20 | +2 | +4 | +9 | -51 |
| 2080s | +77 | +4 | +7 | +18 | -54 |

### 5.1.2    Net precipitation

On an annual basis, net precipitation (precipitation minus evapotranspiration) at the high elevation (1800m) decreased under the 2050s climate, and partially recovered under the 2080s climate (Table 5). Future decreases were greater for clearings than forests, particularly on north-facing sites. For the winter, net precipitation increased under the 2080s climate. For the summer, net precipitation decreased substantially under the 2050s climate, but partially recovered in the 2080s (except for north-facing clearings).



**Table 5: Mean of modelled net precipitation (i.e., precipitation minus evapotranspiration) (mm) by slope aspect, vegetation type, season, and climate condition (CSIRO85) for the high elevation (1800m) (incremental [absolute] change compared to the current climate is provided in parentheses). The south- and north-facing forested sites are moderate density lodgepole pine and spruce-fir stands, respectively. These are the most common stand types for each slope aspect at this elevation.**

| Season | Climate condition | South | | North | |
|---|---|---|---|---|---|
| | | Clearing | Forest | Clearing | Forest |
| Annual | Current | 535 | 408 | 605 | 476 |
| | 2050s | 418 (-117) | 325 (-83) | 476 (-129) | 369 (-107) |
| | 2080s | 452 (-83) | 357 (-51) | 500 (-106) | 414 (-63) |
| Winter | Current | 190 | 174 | 190 | 181 |
| | 2050s | 191 (+1) | 173 (-1) | 191 (+1) | 182 (+1) |
| | 2080s | 220 (+30) | 198 (+24) | 220 (+30) | 209 (+28) |
| Summer | Current | -54 | -102 | -15 | -78 |
| | 2050s | -164 (-110) | -168 (-67) | -145 (-130) | -178 (-100) |
| | 2080s | -150 (-96) | -143 (-42) | -153 (-138) | -151 (-73) |

### 5.1.3 Snowpack accumulation

#### 5.1.3.1 Influence of topography and vegetation under current climate

In the model, snowpack accumulation under the current climate varied directly with elevation and inversely with solar exposure (Fig. 5a & 6a). Clearings generally accumulated more snow than forests, which is a manifestation of the disturbance effect that includes influences on both snowpack accumulation and melt. From the site scale snowpack analysis, the difference in annual maximum SWE between forests and clearings ranged from 4 to 87 mm (2% and 32% lower in forests, respectively) (Fig. 5b). The disturbance effect was greater for higher stand densities (as represented through LAI), and on sites with high solar exposure. From the catchment scale snowpack analysis, the greatest disturbance effect was 123 mm (35% lower in forest), which was in a southerly facing high density Pl stand in the ESSFL zone (~1660 m elevation) (Fig. 6d).

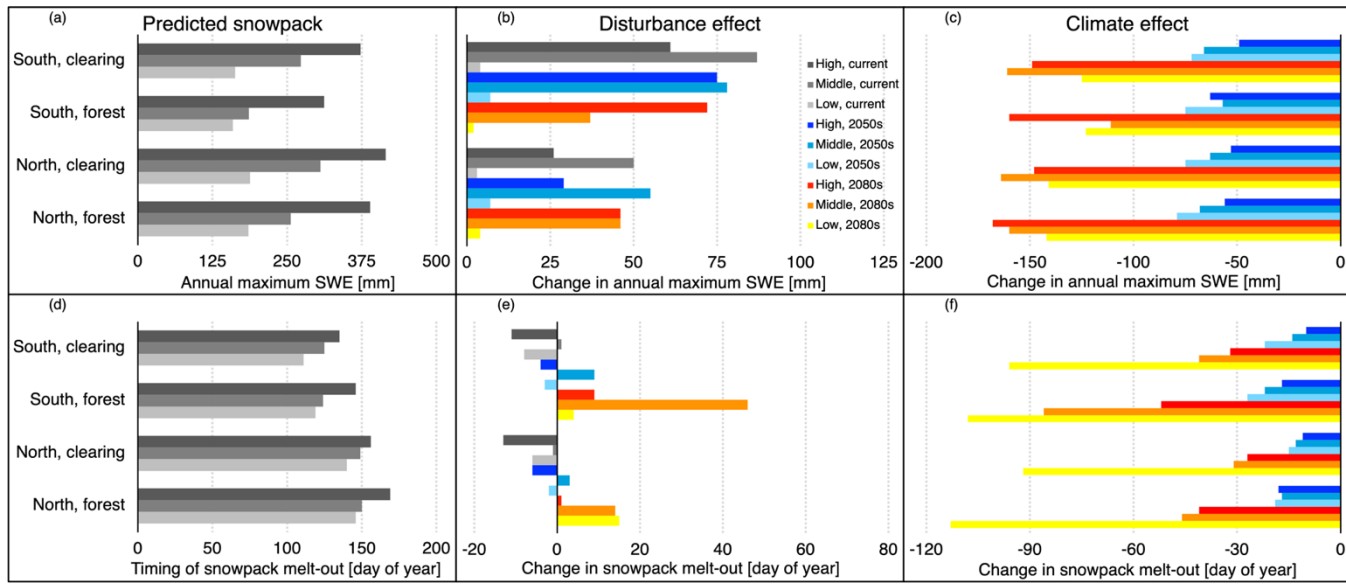

**Figure 5: Mean of annual maximum SWE (mm) and median timing of snowpack melt-out (day of year) by slope aspect, stand type, and elevation for the current climate (panels 'a' and 'd'); and the incremental (absolute) change predicted for the disturbance effect (panels 'b' and 'e') and the climate effect (CSIRO85) (panels 'c' and 'f') (output from site scale snowpack sensitivity analysis). The most common stand type is represented for each slope aspect and elevation, including low density P/F at the low elevation, moderate density P/F at the middle elevation, and moderate density P/F (south aspect) and S/B (north aspect) at the high elevation.**






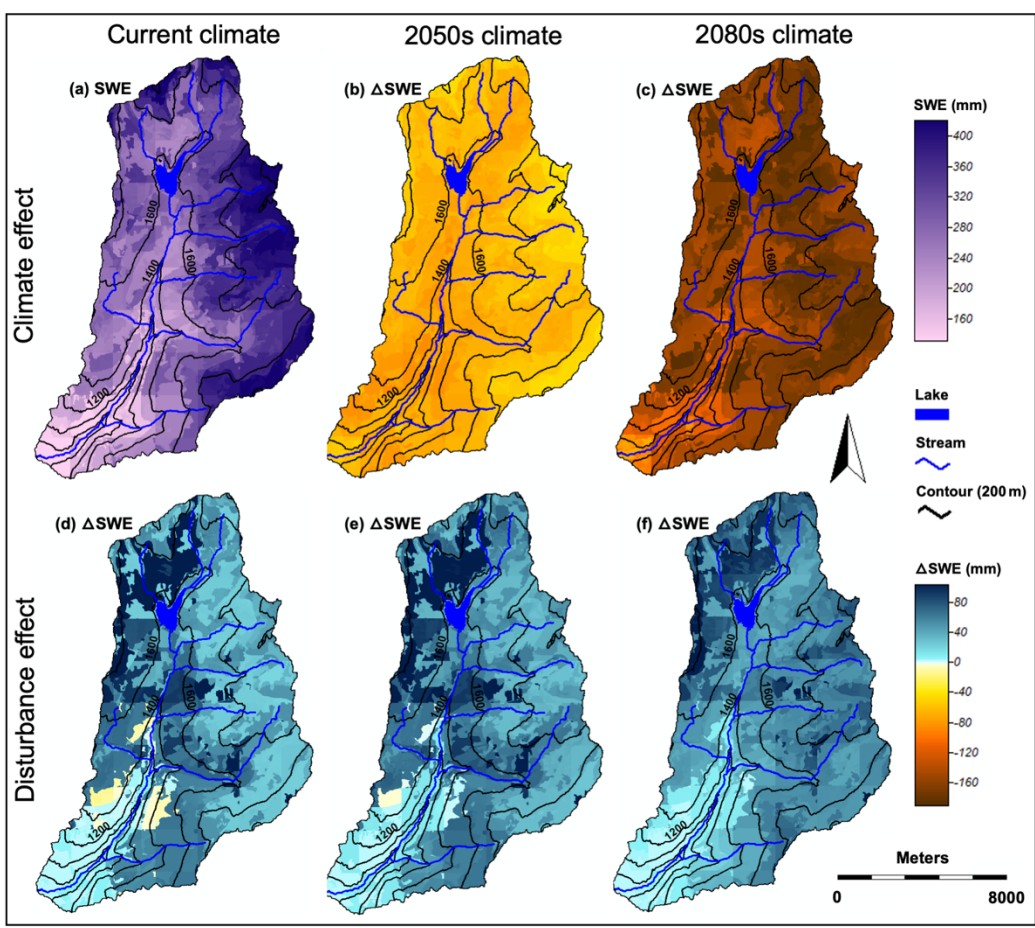

**Figure 6: Distribution of annual maximum SWE for the mature forest condition (panel 'a'), and the absolute change in annual maximum SWE for the mature forest condition under the 2050s (panel 'b') and 2080s (panel 'c') climates (i.e., climate effect) (CSIRO85). For each climate condition, the corresponding disturbance effect (panels 'd'-'f') shows the difference in annual maximum SWE for a 100% cleared condition compared to the mature forest condition.**

### 5.1.3.2   Combined influences of climate and landcover change

Combining all site conditions simulated in the site scale snowpack analysis, the overall mean of the annual maximum SWE was predicted to decrease by 65 mm (-24%) and 146 mm (-55%) under the 2050s and 2080s climates, compared to the current climate, respectively (i.e., climate effect). Decreases were predicted for all site conditions, regardless of elevation, solar exposure, or vegetation type (Fig. 5c & 6b-c). The climate effect was generally greater at the low elevation for the 2050s, but

greater at the middle and high elevations for the 2080s (Fig. 5c).





Clearings generally accumulated more snow than forests under the future climate conditions, for all elevations. Generally, the disturbance effect increased at the high elevation under a changing climate, and decreased at the middle elevation (Fig. 5b & 6d-f). The greatest disturbance effect in the 2050s was on south-facing sites at the middle and high elevations, and on south-facing sites at the high elevation in the 2080s (Fig. 5b).

### 5.1.4    Snowmelt timing

#### 5.1.4.1  Influence of topography and vegetation under current climate

The timing of snowpack melt-out under the current climate varied directly with elevation, and inversely with solar exposure (Fig. 5d & 7a). Sites with high solar exposure melted out earlier than sites with low solar exposure, ranging from 27-29 days earlier at the low elevation, to 14-21 days earlier at the high elevation (larger values correspond to forests). Clearings generally melted out earlier than the corresponding mature forests (i.e., disturbance effect). In the site scale analysis, clearings melted out 6-8 days earlier at the low elevation, and 11-13 days earlier at the high elevation (Fig. 5e). This disturbance effect was minimal at the middle elevation for the vegetation types represented in the site scale analysis (i.e., moderate density P/F; Fig. 5e); however, based on the catchment scale analysis, large disturbance effects were associated with high density stands located in the MS zone at the middle elevation (Fig. 7d).





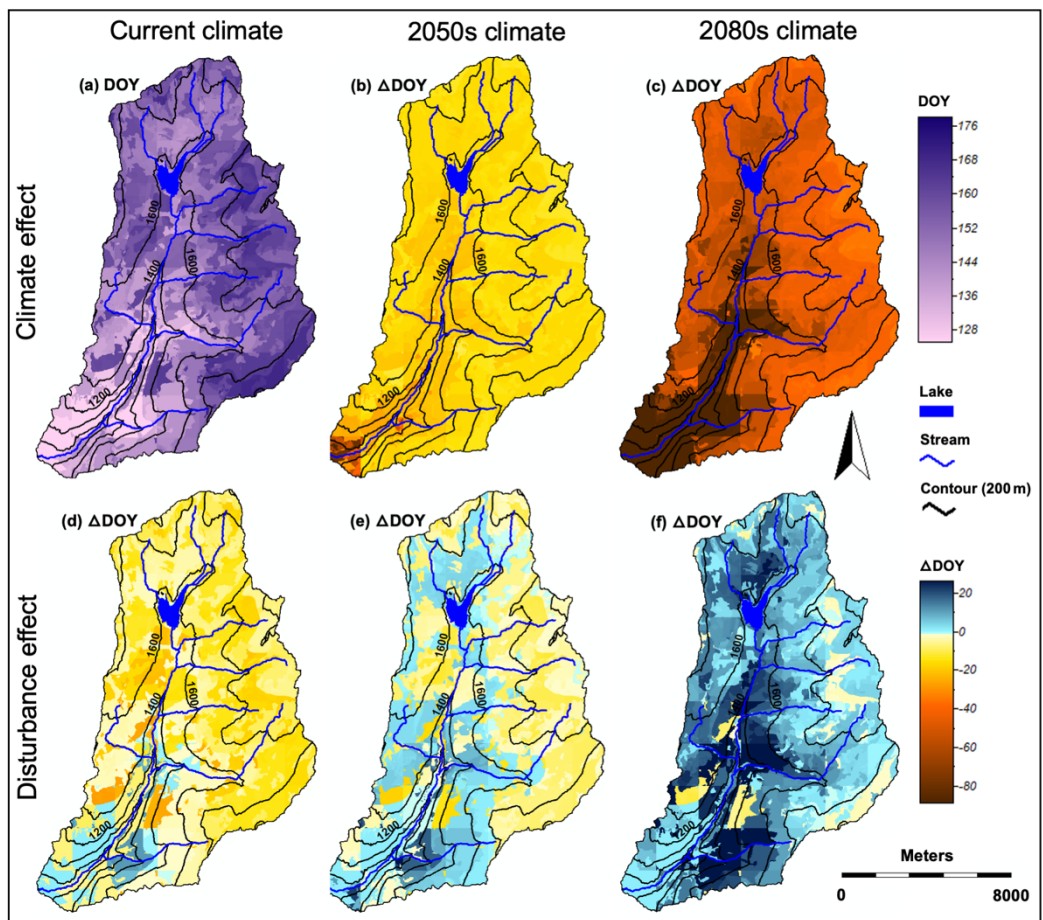

**Figure 7: Distribution of melt-out timing (day of year, DOY) for the mature forest condition (panel 'a'), and the absolute change in melt-out timing for the mature forest condition under the 2050s (panel 'b') and 2080s (panel 'c') climates (i.e., climate effect) (CSIRO85). For each climate condition, the corresponding disturbance effect (panels 'd'-'f') shows the difference in melt-out timing for a 100% cleared condition compared to the mature forest condition.**

#### 5.1.4.2 Combined influences of climate and landcover change

With each successive climate condition, the timing of peak snowpack accumulation advanced, and the duration of the melt phase was extended due to an increase in the persistence of midwinter melt caused by higher air temperatures (Fig. S4). These changes were more persistent for the 2080s climate, with an associated upward shift in the elevations experiencing midwinter melt.

Combining all site conditions simulated in the site scale snowpack analysis, the overall mean timing of snowpack melt-out was predicted to advance by 17 days under the 2050s climate, and 64 days under the 2080s climate (i.e., climate effect). The timing was predicted to advance for all site conditions, regardless of elevation, solar exposure, or vegetation (Fig. 5f & 7b-c).



The climate effect was greatest for lower elevations and higher stand densities (compare Fig. 7b-c to Fig. 2d). In the 2080s, high solar sites generally experienced a larger climate effect than low solar sites (Fig. 5f).


The disturbance effect (i.e., comparing forests and clearings) generally shifted under climate change from advancing to delaying the timing of melt-out (Fig. 7d-f), particularly for south-facing sites at the middle and high elevations (Fig. 5e). This response caused clearings to melt out later than forests under the 2080s climate.

## 5.2    Runoff dynamics

### 540    5.2.1    Timing

For the forested landcover condition, the mean date of the peak flow under CSIRO85 advanced by 18 days and 37 days in the 2050s and 2080s, respectively (i.e., climate effect). Comparing the large burn and forested landcover conditions for the future climates (i.e., disturbance effect), the peak flow advanced by 3 days and 0 days in the 2050s and 2080s, respectively. Thus, the large burn disturbance effect diminished under a changing climate with respect to peak flow timing, as well as hydrograph 545    elongation (Fig. 8). Changes in hydrograph timing were smaller for the other landcover conditions, compared to the large burn.

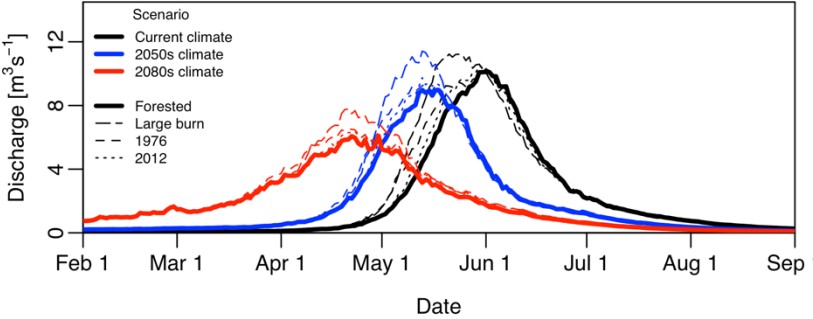

**Figure 8:  Median of daily discharge from the long-term simulations for several landcover conditions under the current and future climates (CSIRO85) (small burn condition excluded for clarity, as it coincides closely with the forested condition).**


### 5.2.2    Peak flow

For all landcover conditions, the peak flow under CSIRO85 decreased with a changing climate for frequently occurring peak flows (<5-10 years for the 2050s; <20-25 years for the 2080s), but increased for extreme events (i.e., climate effect) (Fig. 9b & 10c). The climate effect for the forested condition was a decrease of 12% and 27% in the 2050s and 2080s for events with 555    a 2 year return period (i.e., 50% probability of occurrence in a given year), and an increase of 18% and 22% for events with a 100 year return period (i.e., 1% probability of occurrence in a given year), respectively. Expressed as a frequency shift, a 22.9





m³/s peak flow would have a return period of 100, 33, and 39 years under the current, 2050s, and 2080s climates for the forested condition, respectively. A 22.9 m³/s peak flow would have a return period of 16, 11, and 20 years under successive climate conditions for the large burn (i.e., same large burn landcover condition under three different climate conditions).


Compared to the forested condition, the peak flow increased for all disturbed landcover conditions (i.e., disturbance effect), particularly for extreme events, and regardless of climate (Fig. 9a-b & 10a-b; Table S4). For 100 year events, the large burn disturbance effect was a peak flow increase of 18%, 15%, and 11% under successive climate conditions. Similarly, the disturbance effect for 100 year events was an increase of 6%, 9%, and 9% for the 1976 landcover condition, and 1%, 4%, and

3% for the 2012 landcover condition, respectively. Changes were less than 1% for the small burn.

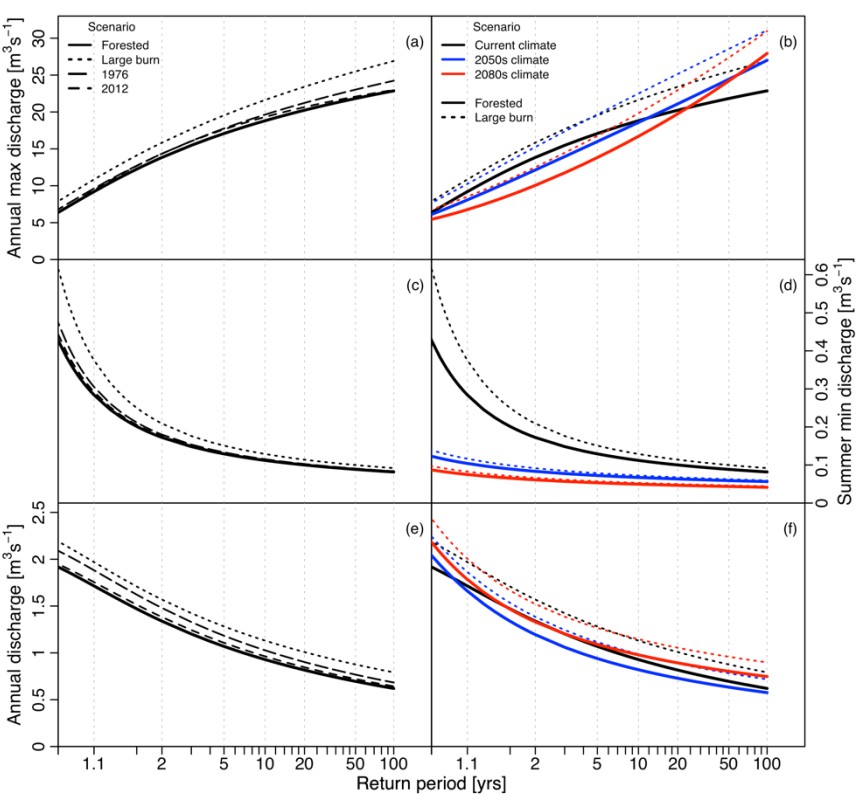

**Figure 9: Event frequency curves from the long-term simulations for the current climate (panels 'a', 'c' & 'e'), and for the forested and large burn landcover conditions under the current and future climates (CSIRO85) (panels 'b', 'd' & 'f'). Small burn condition**
**excluded for clarity (coincides closely with the forested condition).**





**Figure 10: Discharge for 2 and 100 year return periods by landcover condition and event frequency for the current climate (i.e., predicted discharge) (panels 'a', 'd' & 'g'), and the incremental (absolute) change predicted for the disturbance effect (panels 'b', 'e' & 'h') and the climate effect (CSIRO85) (panels 'c', 'f' & 'I'). Values estimated through the event frequency analyses on output from the long-term simulations (Fig. 9). Small burn condition excluded for conciseness (coincides closely with the forested condition).**

### 5.2.3 Summer low flow

For all landcover conditions, summer low flows under CSIRO85 decreased substantially with a changing climate for all return periods (Fig. 9d & 10f). The climate effect for the forested condition was a decrease of 52% and 65% in the 2050s and 2080s for 2 year events, and a decrease of 31% and 49% for 100 year events, respectively. Expressed as a frequency shift, a 0.092 m$^3$/s low flow would have a return period of 100, 2.0, and 1.0 years under successive climate conditions for the large burn condition, respectively. A 0.092 m$^3$/s low flow would have a return period of 37, 1.5, and 1.0 years under successive climate conditions for the forested condition.





For all climate conditions, the disturbance effect increased summer low flows across all return periods for the large burn condition, with intermediary increases for the 1976 condition (Fig. 9c-d & 10d-e; Table S4). Changes were small for the 2012 and small burn landcover conditions. For 2 year events, the large burn disturbance effect was a low flow increase of 21%, 10%, and 8% under successive climate conditions. The disturbance effect for 2 year events was an increase of 3-4% for the 1976 condition, and 1-2% for the 2012 and small burn conditions, regardless of climate.

### 5.2.4 Annual discharge

For all landcover conditions, annual discharge under CSIRO85 decreased with the 2050s climate (compared to the current climate) for all return periods, but fully recovered with the 2080s climate for almost all return periods (Fig. 9f & 10i). The climate effect for the forested condition was a decrease of 11% and 1% in the 2050s and 2080s for 2 year events, and a change of -7% and +20% for 100 year events, respectively. Expressed as a frequency shift for low annual discharge, a 0.79 $m^3$/s annual discharge would have a return period of 100, 46, and >100 years under successive climate conditions for the large burn condition, respectively. A 0.79 $m^3$/s annual discharge would have a return period of 24, 12, and 57 years under successive climate conditions for the forested condition (i.e., low annual discharge is more frequent under the forested condition).

For all climate conditions, the disturbance effect increased annual discharge across all return periods for the large burn condition, with intermediary increases for the 1976 condition (Fig. 9e-f & 10g-h; Table S4). Changes were small for the 2012 and small burn landcover conditions. For 2 year events, the large burn disturbance effect was an annual discharge increase of 17%, 16%, and 14% under successive climate conditions. The disturbance effect for 2 year events was an increase of 10-11% for the 1976 condition, 3% for the 2012 condition, and less than 1% for the small burn, regardless of climate.

## 5.3 Conditions during extreme events

### 5.3.1 Extreme peak flow

Figure 11 shows hydrometeorological conditions for the peak flow event corresponding to the highest discharge in the simulated record for each climate condition under CSIRO85. The peak flow increased under successive climate conditions, with values for the forested condition of 25.2, 28.1, and 30.2 $m^3$/s under the current, 2050s, and 2080s climates, respectively (long-term mean under the current climate was 14.0 $m^3$/s) (Fig. 11g-i). The large burn disturbance effect decreased under successive climate conditions, with corresponding values of +4.1, +3.4, and +1.8 $m^3$/s. During the 4-6 days prior to and during these peak flow events, precipitation generally increased with each successive climate condition, particularly for the 2080s (Fig. 11a-c). Daily low temperatures at the P1 weather station (1619 m elevation) generally remained above 0°C, indicating that most or all precipitation within the catchment fell as rain for the events. The snowpack loading generally decreased and the extent of snow-free terrain during the peak flow increased with each successive climate condition (inferred from the data) (Fig. 11d-f).





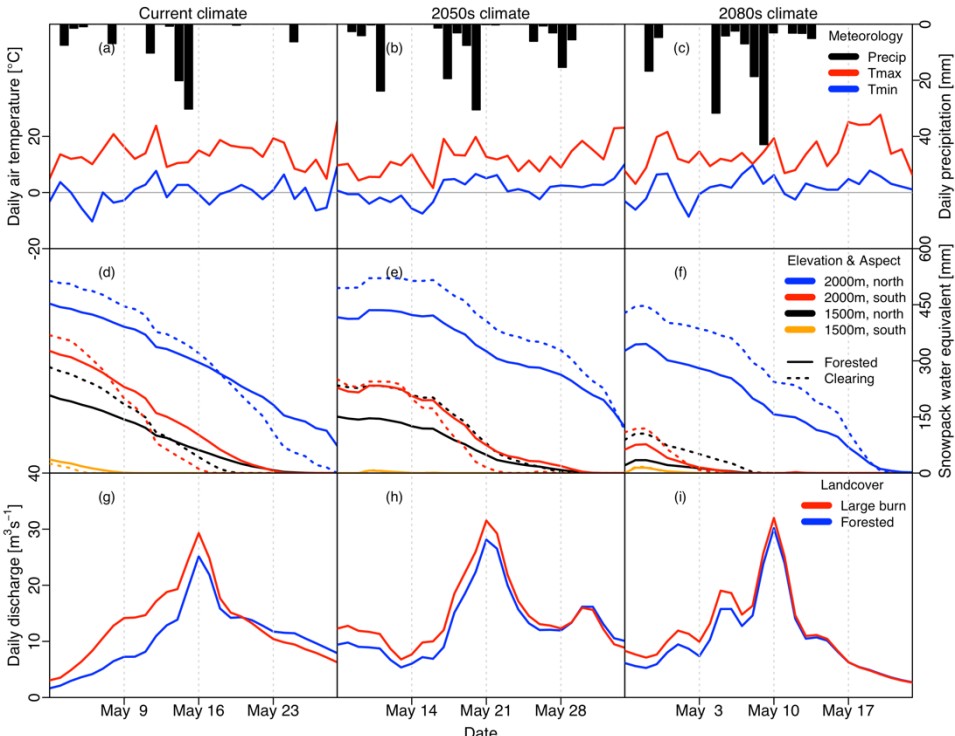

**Figure 11: Hydrometeorological conditions (daily data) for the peak flow event corresponding to the highest discharge in the simulated record for each climate condition (CSIRO85). For the SWE data (i.e., panels 'd' through 'f') solid lines are clearings, 2000m dashed lines are moderate density spruce-fir stands, and 1500m dashed lines are moderate density lodgepole pine stands. These are the most common stand types for each combination of elevation and aspect.**

### 5.3.2 Extreme summer low flow & annual discharge

Figure 12 shows hydrometeorological conditions for the water year corresponding to the lowest summer low flow (Sect. 2.6) and the lowest annual discharge in the simulated record for each climate condition. For the current climate, the same year in the record is represented for summer low flow and annual discharge, whereas different years are represented between summer low flow and annual discharge for the 2050s and 2080s climates. The extreme summer low flow decreased under successive climate conditions, with values for the forested condition of 0.076, 0.055, and 0.040 $m^3$/s under the current, 2050s, and 2080s climates, respectively (long-term mean under the current climate was 0.19 $m^3$/s) (Fig. 12d). The annual discharge values were 0.57, 0.42, and 0.70 $m^3$/s, respectively (long-term mean under the current climate was 1.32 $m^3$/s) (Fig. 12h).

Prior to the extreme summer low flow events for the 2050s and 2080s climates, net precipitation in the winter and early spring was generally higher, and much lower in May and somewhat lower in June, compared to net precipitation during the lowest annual discharge events. Snowpack accumulation was normal or somewhat low prior to the summer low flow events for the 2050s and 2080s, but melted early; whereas snowpack accumulation was well below normal during the annual discharge




events. The conditions prior to the summer low flow events generated peak flow discharge that was normal, but peaked early causing the 2050s and 2080s recession flow to occur approximately one and two months earlier than normal, respectively. In contrast, the conditions during the annual discharge events generated peak flows that were well below normal.

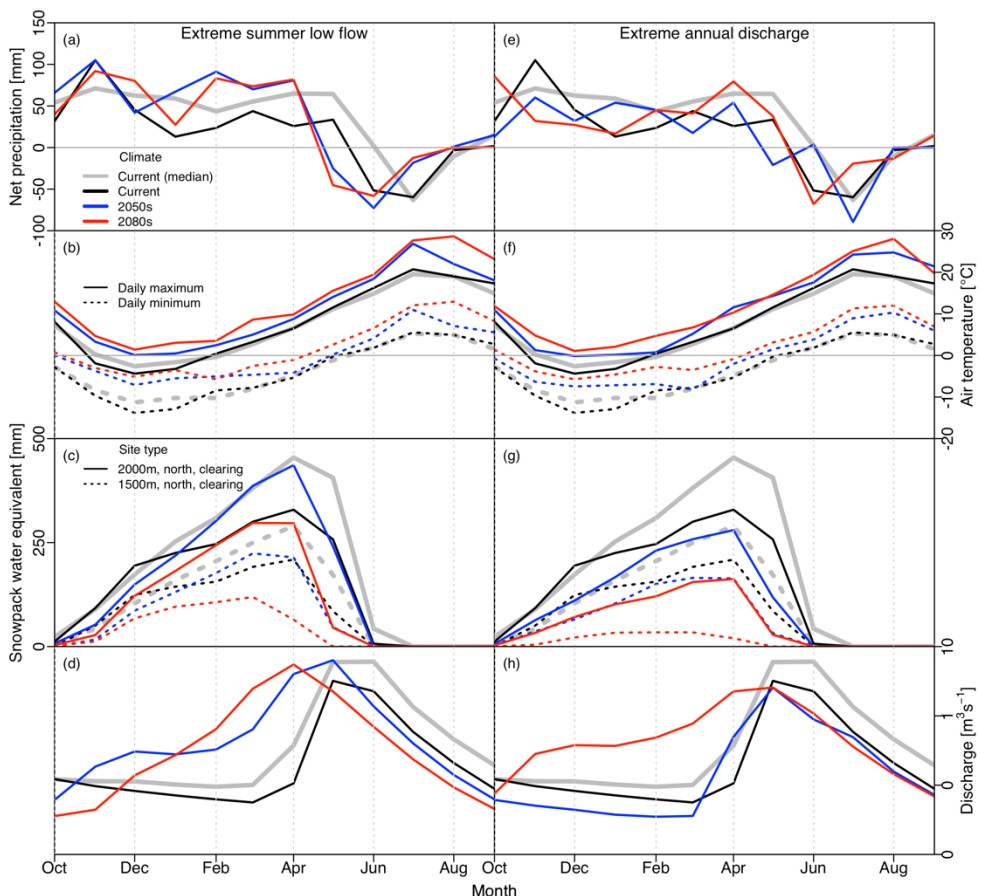

**Figure 12: Hydrometeorological conditions (monthly data) for the water year corresponding to the lowest summer low flow (Sect. 2.6) (panels 'a' to 'd') and the lowest annual discharge (panels 'e' to 'h') in the simulated record for each climate condition (CSIRO85). Panels 'a' to 'c' and 'e' to 'g' show synthetic meteorological conditions at the P1 station. Panels 'd' and 'h' show simulated discharge for the forested landcover condition. The solid grey lines provide the long-term median values for each month under the current climate as a baseline for comparison. Note the logarithmic y-axis in panels 'd' and 'h'.**

## 6    Discussion

### 6.1    Implications to flooding

The finding that stand replacing disturbance can increase peak flows for a period of time after disturbance is well established in the scientific literature (Goeking and Tarboton, 2020). In contrast, the modelling in the current study predicts that the

combined impacts on peak flows of stand replacing disturbance and climate change are generally offsetting for events with
return periods less than 5-25 years (i.e., disturbance increases yield, climate change decreases yield), but additive for extreme
events (i.e., disturbance and climate change both increase yield) (Sect. 5.2.2). CNRM45 in the 2080s is an exception to the
former in that disturbance and climate change were additive for frequent events, likely because of substantial precipitation
increases combined with relatively moderate temperature increases (compared to most other pathways) (Sect. 4). Stands in
advanced stages of regeneration (i.e., 2012 condition) were found to be highly effective at mitigating the influences of
disturbance on peak flows.

For the nearby Upper Similkameen River Watershed, Li et al. (2018) found landcover disturbance and climate variability to
have offsetting influences on the volume of annual surface runoff (i.e., quick flow only; excludes baseflow) and annual
streamflow (i.e., total flow), averaged over a 14 year period (for comparison, a long-term mean flow typically has a return
period of approximately 2 years). Offsetting influences were also found for other watersheds in the BC Interior (Wei and
Zhang, 2010; Zhang and Wei, 2012). However, Giles-Hansen et al. (2019) found landcover disturbance and climate variability
to have additive influences on discharge for the nearby Deadman River Watershed. They suggested that the additive influence
might be associated with the disturbance period being relatively wet (i.e., above normal flows). The findings from these studies
corroborate the current study with respect to a shift from generally offsetting to additive influences for increasing event
magnitude.

The results discussed above for the current study are generated by a complex interplay between landcover change and climate
change with respect to impacts on canopy interception processes, snowpack dynamics, and the resulting runoff regime. This
interplay is interwoven with non-linear runoff response behaviour influenced by changing precipitation intensities, rain-snow
partitioning, and the distribution of forest cover density. This complexity is discussed below.

### 6.1.1  Influence of landcover distribution

The flood frequency results highlight a dependency of extreme peak flows on the distribution of landcover disturbance rather
than strictly the amount of disturbance. This finding is supported by the much greater disturbance effect on peak flows under
the current climate for the large burn compared to the 1976 condition (both had disturbance covering ~1/3 of the catchment)
(Sect. 5.2.2). Catchment SWE increases under both landcover conditions, with greater increases for the large burn due to the
disturbance being at higher elevations where snowpack accumulation and stand densities are greater (Fig. 2d). A larger increase
in catchment SWE for the large burn (Sect. 5.1.3.1) helps explain the greater disturbance effect on peak flows; however,
changes in the synchronization of runoff timing between higher and lower elevations likely also played a role. In this respect,
a post-disturbance advance in the timing of snowpack melt-out in the lower and middle elevations (i.e., early melting areas)
(1976) (Sect. 5.1.4.1) generates antecedent conditions with drier soils and less snowpack during the peak flow period, and,
thus, results in a lower contribution to the peak flow (i.e., desynchronization). In contrast, an advance in the timing of snowmelt





at higher elevations (i.e., late melting areas) (large burn) synchronizes the timing of runoff with downslope areas during the peak flow. Increased synchronization could increase the peak flow over and above that caused by strictly increasing catchment SWE, and vice versa for desynchronization (Bewley et al., 2010; Ellis et al., 2013; Pomeroy et al., 2012; Winkler et al., 2015).


A comparison between the large burn, 1976, and 2012 conditions further highlights the influence of landcover distribution on runoff, whereby a catchment disturbance level of 47% (2012) generated little or no disturbance effect on peak flows under the current climate (Sect. 5.2.2). The 2012 condition incorporated forest stands in various stages of regeneration (Fig. 3e). The catchment-wide LAI was 92% as high as that for the forested condition, versus values of 66% and 83% for the large burn and

1976 conditions (Table 2). Thus, the 2012 condition illustrates hydrologic recovery of the disturbance effect through recovery in the rates of evapotranspiration and, likely, the snowmelt energy budget. For UPC and Mayson Lake, Winkler and Boon (2015) showed ~80% recovery in the disturbance effect on snowpack ablation and accumulation for regenerating lodgepole pine stands at approximately 50-60% of the mature height. Also for UPC, Winkler et al. (2021) showed more-or-less full recovery of the soil moisture content in the root zone of regenerating lodgepole pine stands at approximately 10% of the mature

height.

### 6.1.2    Influence of climate change

The results illustrate an increasing importance of rainfall in controlling peak flow response under a changing climate, at the expense of snowmelt influence (Sect. 5.3.1) (Merritt et al., 2006; Whitfield et al., 2002). This finding is supported by three points: (1) the increasing persistence of midwinter rainfall and snowmelt; (2) the decreasing magnitude of frequently occurring

peak flows that correspond to decreasing snowpack loads; and (3) the increasing magnitude of extreme peak flows associated with increasing spring rainfall intensity. The latter outcome occurred even with an increase in the extent of snow-free terrain during the peak flow, further highlighting the increasing influence of rainfall in controlling runoff (e.g., rain on snow, rain on wet soils). These findings are based primarily on the CSIRO85 results, but the first two points are generally consistent with the hydrograph changes for most of the emission pathways (CNRM45 in the 2080s being an exception).

### 6.1.3    Combined influences of landcover and climate

The combination of climate change and the large burn condition is predicted to advance the timing of the peak flow by two to nine times (three to five times for CSIRO85) more than the advance generated by the large burn condition alone (Sect. 4 & 5.2.1); thus, showing a dominant effect of climate on the timing of spring freshet. With respect to the magnitude of extreme peak flows, the CSIRO85 results show a similar influence for landcover disturbance and climate change. However, the results

also suggest a decrease in the sensitivity of extreme peak flows to disturbance at high elevations under a changing climate (Sect. 5.2.2). This decreasing sensitivity is demonstrated by the decrease in the large burn disturbance effect with successive climates. Similarly, Cristea et al. (2013) found the climate effect on the timing of spring freshet runoff and peak flow yield to be greater for a forested catchment condition (dense lodgepole pine in lower 39% of basin, with alpine above) than a disturbed


condition. The current study also suggests that, as rainfall runoff becomes more important for extreme peak flows under a
changing climate, the specific landcover condition in the middle and low elevations will also become more important. This
finding is supported by the increasing disturbance effect for the 1976 condition with successive climates (Sect. 5.2.2).

We inferred that the decreasing sensitivity to the large burn condition under climate change is related somewhat to the canopy
interception capacity becoming more overwhelmed by the increasing rainfall intensity during extreme events (i.e., lower
percentage of rainfall intercepted) (Sect. 5.1.1) (Spittlehouse and Maloney, 2023). In this respect, Asano et al. (2020) found
that the peak propagation speed of hillslope runoff increased by two orders of magnitude with increasing rainstorm size. Lewis
et al. (2001) found that logging-induced relative increases in peak discharge declined with increasing event magnitude.
Moreover, we inferred that the increasing sensitivity to the 1976 condition under climate change is likely related to increasing
catchment-wide synchronization of runoff caused by decreasing snowmelt inputs (asynchronous distribution throughout the
catchment) being replaced by increasing rainfall inputs (relatively even distribution throughout the catchment). The snowpack
results offer additional insight into these changing sensitivities. That is, the modelling predicts that clearings will moderate the
impacts of climate change on snowpack accumulation at the high elevation, whereas this influence is opposite or negligible at
the middle and low elevations (Sect. 5.1.3.2).

## 6.2    Implications to water supply

### 6.2.1    Summer low flow

Similar to the conclusions of Dierauer et al. (2021), our results suggest that extreme summer low flows will become
commonplace in the future, with most of the change in frequency occurring by the 2050s (Sect. 5.2.3). These findings are
based primarily on the CSIRO85 results, but are corroborated by August discharge for all emission pathways (Fig. 4). With
respect to controls on the severity of extreme summer low flows, the hydrometeorology conditions shown in Figures 11c and
11d suggest that the timing of snowmelt and the post-freshet recession flow are more important than the volume of snowmelt
and spring freshet runoff. An advance in the timing of spring freshet runoff would cause an earlier start to the low flow period
and provide additional time for soil desiccation during the summer, which would decrease the low flow (Dierauer et al., 2021).
Notwithstanding this influence, summer low flows were increased by the large burn under the current climate, even while
generating a small advance in the timing of the recession flow (Fig. 8). This contrast suggests that disturbance related lower
ET (increases low flow) and earlier melt-out (decreases low flow) have unequal impacts, with lower ET having a greater
influence on the severity of low flows.

The large burn effect on the timing of the peak flow and recession flow transitions from a one week advance under the current
climate, to a smaller advance or a small delay under the 2080s climate, depending on emission pathway (Fig. 4 & 8). The
CSIRO85 results indicate that this reversal in the disturbance effect is associated most directly with disturbance to high density





stands on a southerly exposure at the middle and high elevations (Sect. 5.1.3.2 & 5.1.4.2). We infer that this reversal is related primarily to meteorological related changes within forests – likely decreasing snowpack accumulation and increasing below canopy net long-wave radiation associated with increasing air temperature (Cristea et al., 2013). These conditions would generate more frequent midwinter melt in forests. Notwithstanding these findings, the decreased sensitivity of the low flow

volume to landcover condition (with up to 1/3rd of a catchment disturbed) under a changing climate suggests that the influence of stand replacing disturbance will likely decrease. To exemplify this point, the large burn increased the 2 year summer low flow by 0.037 $m^3$/s under the current climate, whereas the increase was only 0.0051 $m^3$/s with the 2080s climate under CSIRO85 (Fig. 10e).

### 6.2.2    Annual discharge

The CSIRO85 results indicate that low annual discharge is predicted to become more prevalent by the 2050s, but then fully recover or become less prevalent (compared to the current climate) by the 2080s because of increased precipitation in the fall-spring period (Table 3) (Rasouli et al., 2014). Rasouli et al. (2019) found that reduced snowpack sublimation and increased precipitation offset the impact of increased summer ET, causing annual yield to be maintained under a changing climate. The current study also suggests that landcover disturbance (e.g., large burn) can have a mitigative influence on low water supply

that is predicted to be sustained under a changing climate for annual discharge, but minimally for low flow (Fig. 10e & 10h). A low annual discharge occurring, on average, once in a 100 year period under the large burn condition, would occur four to five times under the forested condition for the current climate or either future climate (CSIRO85) (Fig. 9f). The mitigative influence is generated primarily by increased spring freshet runoff related to reduced winter ET – an influence that diminishes as forest stands reach advanced stages of regeneration (e.g., 2012 condition) (Fig. 9e).


For the Penticton Creek Watershed, the snowpack analyses suggest that the largest disturbance related increases in annual runoff yield likely occur with high density stands at the middle and high elevations, particularly high density stands in the high elevation with a southerly exposure (Sect. 5.1.3). This response is predicted to increase under a changing climate for the high elevation, whereas the snowpack and runoff analyses (Sect. 5.2.4) suggest that the response to disturbance in the low and

middle elevations will not vary substantially with climate. We were unable to find existing literature to compare to these findings.

Winkler et al. (2017) found that increases in annual yield within the 241 Creek sub-catchment averaged only 5% after 47% of the catchment was logged, compared to an increase of 17% for the large burn in the current study for a 2 year event under the

current climate. Possible reasons for this discrepancy include large differences in catchment scale, and differences in the distribution of disturbance relative to stand density. In particular, the highest density stands in the Penticton Creek Watershed, which showed the greatest disturbance effects, do not generally coincide with the 241 Creek sub-catchment (compare Fig. 1, 2d, and 6d). The original stands in the 241 Creek sub-catchment had pre-dominantly moderate densities (Fig. 2d).



### 6.2.3    Interplay between low flow & annual discharge

We infer that soil moisture retention from snowmelt and spring rainfall is the primary water source for summer baseflow and ET. Thus, advances in the timing of snowmelt and the recession flow (Fig. 8) provide a partial explanation for why climate change is predicted to decrease summer low flow severely, but not annual discharge (Cristea et al., 2013). The earlier snowmelt and recession flow also explain why landcover disturbance can increase annual discharge without increasing low flow. That is, increased spring freshet yield from landcover disturbance (i.e., reduced winter ET) does little to increase low flow if the

timing of snowmelt and the recession flow are advanced substantially by climate change (Sect. 5.3.2 & 6.2.1). For comparison, Mirmasoudi et al. (2019) found climate change increased total spring water supply by 35-39%, and decreased summer water supply by 36-79%. Dierauer et al. (2018) found that warm winters (consistent with an earlier recession flow) correspond to longer, more severe summer low flows.

Another reason for the greater influence of climate change on summer low flow than annual discharge is likely a decrease in late spring and summer net precipitation (Table 5; Fig. 12a), driven by increasing air temperature combined with changes in the seasonal distribution of precipitation (Table 3). Hale et al. (2022) found that a warming related shift from spring snowmelt runoff toward winter rainfall runoff resulted in an increase in annual yield, even without a change in the seasonal distribution or amount of precipitation, but also increased rates of summertime drying.

### 6.3    Managing watershed risk

When approaching watershed management from a hydrologic risk evaluation / risk management perspective, the study results indicate there is a need to carefully evaluate the interplay among environmental variables, the landscape, and the values at risk. There is potential for the same management strategy to cause offsetting and additive influences on different risks. For instance, under the current climate, the large burn (i.e., stand replacing disturbance) increased peak flows (increasing risk), increased

annual discharge (decreasing risk), and increased low flows (decreasing risk). Climate change decreased low flows and the large burn increased low flows, generating offsetting influences; however, climate change increased peak flows for extreme events and the large burn increased peak flows regardless of climate, generating additive influences.

The study results also indicate that management of hydrological risks should take a long-term perspective (e.g., 40-100 years)

in temperate forest ecosystems because of the slow growth of trees, the changing climate, the variable effects of landcover disturbance on hydrology, and their interactions. For instance, in contrast to the large burn, the 2012 landscape showed little hydrological effect from 47% of the catchment experiencing disturbance spanning a 40+ year period (Fig. 10b,e,h). Moreover, the sensitivity of extreme peak flows to stand replacing disturbance was predicted to decrease under a changing climate for disturbance at the high elevation, but increase for disturbance at the low elevation due to the influence of rain (Sect. 6.1.2 &





6.1.3). These latter findings are counter to the current risk management strategy of protecting high elevation forests and harvesting in lower elevation areas.

Having no plan or only a short-term plan for managing landcover is not helpful for managing long-term hydrologic risk. There is potential for uncontrolled landcover disturbance (e.g., extreme wildfire, forest pests) that might (1) increase extreme flooding

from widespread stand replacing disturbance (near-term impact) (Goeking and Tarboton, 2020), (2) decrease annual discharge and summer low flows from non-stand replacing disturbance (near-term impact) (Adams et al., 2012; Biederman et al., 2015; Goeking and Tarboton, 2020), and/or (3) decrease annual discharge and summer low flows when a state of dense immature forest cover is reached many years after widespread stand replacing disturbance (future impact) (Crampe et al., 2021; Perry and Jones, 2017; Segura et al., 2020). Short-term management plans do not allow the full spectrum of hydrologic risks to be

managed holistically.

The study results also show that the spatial distribution of mitigation strategies needs to be considered because synchronization and desynchronization of snowmelt timing can either exacerbate or mitigate peak flow risk. This finding is supported by two points: (1) different peak flow responses to different landcover distributions having similar overall disturbance levels (Sect.

6.1.1); and (2) variation in the disturbance effect on the timing of snowpack melt-out in relation to elevation, solar exposure, and pre-disturbance stand density (Sect. 5.1.4). This finding is supported by other synchronization related studies (Bewley et al., 2010; Ellis et al., 2013; Pomeroy et al., 2012; Winkler et al., 2015; Zhao et al., 2021).

Key pieces of the risk management toolset include wildfire management (suppression, fuel reduction, and prescribed burning)

and forest management (harvesting, forest health treatments, adaptive planting, diversification). However, risk evaluation and management planning should also consider other mitigation options, including the capacity of upland reservoir storage, the volume and timing of water consumption, and municipal infrastructure. In particular, reservoir storage is likely to become more important for maintaining water supply under a changing climate because of increasing extreme peak flows, earlier spring freshet, and more severe summer drought.


The detailed study results are relevant throughout the Thompson-Okanagan Plateau in British Columbia, which has an area of 32,000 km$^2$ and provides water to a population of approximately 700,000. However, we believe the key findings can be generalized globally to other snowmelt-dominated montane catchments having variable slope aspects and substantial elevation relief (e.g., >500 m), particularly those with forest cover disturbance (harvesting or wildfire) and a managed water supply.

**6.4    Uncertainty**

The modelling did not incorporate an uncertainty analysis addressing model parameterization. In this respect, the study findings rely on the quality of the model fit as it relates to model performance (Sect. 3). In addition, model parameter values were





assumed to remain static with climate change; thus, any long-term climate related changes in catchment processes that would substantially change runoff behaviour could result in different outcomes than those predicted in the study. For this reason, a

greater level of certainty should be placed on the 2050s results than the 2080s results.

With respect to meteorological data, we utilized 100 years of synthetic data for each modelling scenario. We acknowledge that rarer events can occur (e.g., probable maximum flood) that could lie outside the boundaries of our simulation.

A landcover condition with more than $1/3^{rd}$ of the catchment in a recently cleared condition was not considered in this study and, therefore, the potential impacts of more widespread forest cover disturbance are unknown. Moreover, climate and landcover effects on discharge were not analyzed at different catchment scales. It is expected that impacts on event frequency would likely be more severe at smaller scales, and less severe at larger scales, because of space-time integration/averaging of runoff processes.


The potential for wildfire to generate water repellent soil was not considered. The permeability of the soil can be severely diminished by fire, to a state where intense rainfall cannot effectively infiltrate and/or percolate, causing rapid overland flow. Under these conditions, the landcover effect on peak flows could be orders of magnitude greater than predicted in this study, particularly over smaller scales (e.g., sub-catchment or smaller), and within the first few years after fire (DeBano, 2000).

**7      Conclusions**

The combination of climate change and stand replacing landcover disturbance in the middle and high elevations is predicted to advance the timing of the peak flow two to nine times (depending on emission pathway) more than the advance generated by disturbance alone, showing a dominant effect of climate on the timing of spring freshet. The combined impacts of climate change and landcover disturbance on peak flow magnitude are generally offsetting for events with return periods less than 5-

25 years, but additive for more extreme events. There is a dependency of extreme peak flows on the distribution of landcover, where higher elevation disturbance has a greater effect than lower elevation disturbance. This difference is caused by greater increases in snowpack accumulation and higher stand densities at higher elevations, and changes in the synchronization of runoff timing between higher and lower elevations. However, the results also suggest a decrease in the sensitivity of extreme peak flows to high elevation landcover disturbance under a changing climate. Moreover, stands in advanced stages of

regeneration are highly effective at mitigating the influences of disturbance on peak flows.

The modelling predicts an increasing importance of rainfall in controlling peak flow response under a changing climate, at the expense of snowmelt influence. Changes include more frequent midwinter rainfall and snowmelt, decreases in the magnitude of frequently occurring peak flows corresponding to decreasing snowpack loads, and increases in the magnitude of extreme



peak flows associated with increasing spring rainfall. As rainfall runoff becomes more important for extreme peak flows under a changing climate, the specific landcover condition in the middle and low elevations is also predicted to become more important, likely because of increasing catchment-wide synchronization of runoff.

The results suggest that extreme summer low flows will become commonplace in the future, with most of the change in
frequency occurring by the 2050s. The timing of snowmelt and the post-freshet recession flow appear to be more important than the volume of snowmelt and spring freshet runoff for influencing the severity of low flows. Stand replacing landcover disturbance is predicted to increase low flows, but this influence is predicted to diminish under a changing climate.

The occurrence of low annual water yield is also predicted to become more prevalent by the 2050s, but then fully recover or
become less prevalent (compared to the current climate) by the 2080s because of increased precipitation in the fall-spring period. The modelling suggests that stand replacing landcover disturbance increases annual yield under current and future climates. This mitigative influence is generated primarily by increased spring freshet runoff related to reduced winter ET – an influence that diminishes as forest stands reach advanced stages of regeneration.

The study results demonstrate the importance of examining complexity in three dimensions with respect to modelling changes to the hydrological regime: climate change, landcover change (disturbance and regrowth), and numerous hydrological indicators. The indicators include influences on snowpack accumulation and melt; rainfall dynamics; runoff timing, magnitude, and frequency for peak flows, low flows, and annual discharge; and frequent and extreme events. Moreover, for managing watershed risk, the results indicate there is a need to carefully evaluate the interplay among environmental variables, the
landscape, and the values at risk. Strategies to reduce one risk may increase others, or effective strategies may become less effective in the future. Risk management should consider many options, including wildfire management, forest management, upland reservoir storage, water consumption, and municipal infrastructure. Moreover, management of hydrological risks should take a long-term perspective (e.g., 40-100 years).

## 8      Authour contribution

Russell Smith prepared the manuscript with contributions from all co-authours. He developed the Raven catchment model, and designed and implemented the modelling investigations and analyses. Caren Dymond conceptualized the study within an overarching water risk project that was set up for the Penticton Creek Catchment, and secured funding. She also led the LANDIS-II landcover modelling. Dave Spittlehouse led the meteorological monitoring program in the Penticton Creek Watershed, and developed the synthetic weather records for current and projected future conditions. Rita Winkler led the
snowpack and hydrometric monitoring program in the Penticton Creek Watershed. Georg Jost provided an initial Raven model





setup and advised on further development of the model. Caren, Dave, and Rita provided multiple comprehensive reviews of the manuscript.

## 9 Competing interests

The authours declare that they have no conflict of interest.

**10 Acknowledgements**

We would like to thank James Craig and Robert Chlumsky for their assistance with setting up and implementing Raven, as well as organizations that have supported the development of Raven (University of Waterloo, BC Hydro, Canadian Hydraulics Centre). Luke Crevier, Rachel Plewes, and Fergus Stewart provided processed spatial datasets. Barbara Zimonick and Gary Van Emmerik aided in data collection. Craig Nitschke provided TACA model output. Sheena Spencer provided a friendly
review. The City of Penticton provided information on water management in the study catchment. Funding was provided by the B.C. Ministry of Forests. Finally, we would like to thank two anonymous reviewers and the HESS Editor, Markus Weiler, for their constructive review comments.

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
