# Peer review of "Modelling the effects of climate and landcover change on the hydrologic regime of a snowmelt-dominated montane catchment"

_Hydrology and Earth System Sciences, 2024_

## Author Comment (AC1)

**Author response to reviewer #1 comments for HESS manuscript "Modelling the effects of climate and landcover change on the hydrologic regime of a snowmelt-dominated montane catchment" [Paper #: hess-2024-361]**

Dear Reviewer,

We would like to thank you for your thorough and thoughtful review of our manuscript submission. You raised many important issues that will certainly result in a stronger manuscript. Please find below a list of responses to your comments. We hope our responses satisfy the spirit and intent of your remarks.

Sincerely,

Russell Smith

**Reviewer #1 comments**

**General Comments**

This paper combines a large amount of data and different climate and land cover scenarios in a modelling study to determine the combined effects of land cover change and climate change on the snowpack and streamflow regime for a headwater catchment in British Columbia, Canada. This is a huge undertaking and I appreciate that the effects are analysed for many different aspects of the snowpack and hydrograph. The paper clearly shows the interaction between the effects of land cover change and climate change. For some aspects of the hydrograph or snowpack, the land cover change enhances the effects caused by climate change and for others, it mitigates it. The results furthermore highlight the importance of the location of the disturbance in the catchment (i.e., whether the vegetation is replaced in the upper or lower part of the catchment) and the time since the disturbance. These are important results and highlight the need to consider the effects of land cover and climate change jointly and to not study the effects of land cover change for only one climatic period.

- Thank you for the very positive comments, and for acknowledging the large effort.

Unfortunately, some of the model decisions are not so clearly described and it is not very clear how the model was calibrated. There is also no mention of the uncertainties in the results due to parameter uncertainty. Considering the potentially very large number of parameters that are optimized, it is possible that a different parameter set would lead to considerably different results. The lack of uncertainty analyses is acknowledged in the final part of the discussion, but I would argue that at least some model uncertainties need to be presented. As a result of the lack of a clear description of the calibration procedure and the lack of an uncertainty analysis, it is not clear how the presented results are influenced by the model decisions or model parameter sets (equifinality).

- In your comments above and below, you point out several areas in the manuscript where the optimization process could be clarified. We acknowledge the need to provide clarity around the issues you identified, and have addressed your related comments in the sections below. We propose increasing our discussion of uncertainties in the manuscript (Section 6.4) to address parameter uncertainty in greater detail. We will also provide more details in the manuscript from the points below related to the calibration process.

- We further acknowledge the importance of considering uncertainties, including variability in meteorology and landcover (which we addressed thoroughly), and parameter uncertainty. We acknowledge that a comprehensive parameter uncertainty analysis would be an asset to the manuscript; however, it would be a very large undertaking, on top of the very large undertaking we have already completed (as you acknowledged). While your request for a parameter uncertainty analysis may seem like a modest undertaking, it is important to consider the amount of modelling and data synthesis involved. In this respect, consider the multiplication effect of eleven climates, five landcovers, several key hydrologic variables (snowpack accumulation, snowmelt timing, peak flow, annual yield, low flow, extreme events), numerous alternative parameter sets, and 100 years of daily simulation for each combination of these. Additionally, we calibrated the model using the Dynamically Dimensioned Search algorithm (Section 2.3.4 in manuscript). To generate a selection of alternative parameter sets for the uncertainty analysis, we would need to recalibrate the model using a randomized search algorithm (e.g., Monte Carlo simulation). As noted below, we believe the model was constrained well by the calibration process, resulting in limited parameter uncertainty. Furthermore, we do not have the resources (funding or time) available to complete such a parameter uncertainty analysis. It is also important consider that the paper is already long, as you acknowledged. We found it challenging to clearly and effectively present the results that are already provided. Accounting for uncertainty (e.g., error bars, additional lines or plots) would exacerbate that challenge.

- Further to the points above regarding parameter uncertainty, we believe the model was constrained well by the calibration process, resulting in much less parameter uncertainty than is suggested. All model parameters were calibrated simultaneously on the SWE and discharge datasets shown in Figures S2.1 through S2.5 (manuscript supplement) and on annual precipitation at Penticton Airport, then validated on the datasets shown in Figures S3.1 through S3.5. It is a rich optimization dataset that encompasses a large range of:
    - intra-seasonal and inter-seasonal meteorological and hydrological conditions (e.g., 2010 vs. 2011 SWE; 1972 vs. 1973 discharge in Penticton Creek; 1986 vs. 1990 vs. 1992 discharge in 240, 241, and Dennis Creeks, see Figures S2.1 through S2.5),
    - catchment forest cover conditions (e.g., 240 Creek Sub-catchment vs. 241 and Dennis Creek Sub-catchments, see Figures 1 and 2 in main body),
    - catchment elevations and scales (e.g., Penticton Creek Watershed vs. sub-catchments),
    - catchment orientations (e.g., 241 Creek vs. Dennis Creek Sub-catchments),
    - forest cover conditions at SWE stations (e.g., UP13 vs. UP2, UP9 vs. UP10, see Figure S2.5; UP1 and UP3 vs. UP2 and UP4, see Figure S3.5),
    - elevations of SWE stations (e.g., UP13 vs. UP9, 2F08 vs. UP10), and
    - orientations of SWE stations (e.g., UP9 vs. UP13).
- To support the parameterization, a forest cover survey was completed that encompassed seven mature forest plots ranging in elevation from 649 m (dry, low elevation forest) to 1930 m (wet, sub-alpine forest) (locations provided in Figure 1). The sampling protocol was based on Canada's National Forest Inventory Ground Sampling Guidelines (Canadian Forest Inventory Committee, 2008). Measured variables included leaf area index, crown closure, tree diameter, tree height, and species, among others. Leaf area index and crown closure (particularly important in the model) were measured at 19 points per plot using hemispherical photos.
- A composite objective function was utilized (see Table S3 in supplement) that allowed the optimization to be constrained well by focusing on different features of the optimization dataset. Each data feature and component of the objective function were important for different reasons, and provided value for constraining different model parameters. For instance:
    - Overall yield (i.e., absolute bias) and variance (i.e., NSE) were applied separately, to ensure that neither could be fit well at the expense of the other.

- o Absolute bias constrained parameters controlling overall water volume (e.g., precipitation lapse rates, evapotranspiration in winter and summer).
- o NSE constrained parameters controlling the timing of snowmelt and runoff (e.g., energy balance, runoff routing).
- To ensure that internal model processes were functioning well, individual components of the composite objective function were applied separately to discharge and SWE, smoothed and unsmoothed discharge, spring freshet and low flow, and sub-catchments and the main catchment outlet.
- The table below outlines factors that were important for constraining key parameters. The constraining features identified in the table relate to features in the optimization dataset that were particularly important for constraining the corresponding parameter(s).

| Parameter type | Parameter or parameter group | Constraint |
|---|---|---|
| Short-wave radiation | CloudTempRanges | Calibration range was based on a comparison of measured short-wave radiation and diurnal air temperature for P1 climate station.

Constrained by calibration on SWE and discharge.

Constraining features:

• rate of snowmelt and associated runoff volume during intra-seasonal sunny periods versus cloudy periods

• snowmelt timing and associated runoff timing in wet vs. sunny spring freshet seasons

• differences in snowmelt and runoff between different slope aspects / catchment orientations. |
| | UBCCloudPenetration | Calibration range was based on ratio of low mid-day short-wave radiation vs. high mid-day short-wave radiation for P1 climate station.

Constrained by calibration on SWE and discharge.

Constraining features:

• rate of snowmelt and associated runoff volume during sunny periods

• differences in snowmelt and runoff between different slope aspects / catchment orientations. |

| | | |
|---|---|---|
| Air temperature | AdiabaticLapseRate, WetAdiabaticLapse, UBCTempLapseRates | Calibration range was based on lapse rates calculated using weather data from P1 and Penticton Airport stations.

Constrained by calibration on SWE and discharge.

Constraining features:

• differences in snowpack accumulation and snowmelt rates at different elevations during periods when air temperatures are close to zero degrees

• differences in snowpack accumulation and snowmelt rates at different elevations for wet vs. dry periods

• differences in volume and timing of runoff during snowmelt periods for sub-catchments vs. main catchment outlet |
| | ReferenceMaxTemperatureRange | Calibration range was based on diurnal air temperature for P1 climate station during clear sky conditions, and based on values suggested in the user manual for UBC Watershed Model (Quick, 1995).

Constrained by calibration on SWE and discharge.

Constraining features:

• rate of snowmelt and associated runoff volume during intra-seasonal sunny periods versus cloudy periods

• snowmelt timing and associated runoff timing in generally wet vs. sunny spring freshet seasons |
| Precipitation | PrecipitationLapseRate | Calibration range was based on calculation of long-term precipitation lapse rate between Penticton Airport and P1 climate stations.

Constrained by calibration on long-term mean precipitation at Penticton Airport.

Constraining features:

• rate of snowpack accumulation in clearings during snowfall events

• volume of runoff during rainfall events |
| | RainSnowTransition | Values assigned based on those suggested in the user manual for UBC Watershed Model, and based on values used in BC Hydro operational models. |

| Snowpack albedo | UBCSnowParams (P0ALBMIN, P0ALBMAX, P0ALBREC, P0ALBASE, P0ALBSNW, P0ALBMLX) | Calibration ranges based on values in literature (e.g., Spittlehouse and Winkler, 2004) and values used in BC Hydro models. Constrained by calibration on SWE and discharge. Constraining features:
 • differences in rate of snowmelt and associated runoff volume during sunny periods after recent snowfall vs. after several days without snowfall |
|---|---|---|
| Maximum liquid water content of snow | IrreducibleSnowSaturation | Value assigned based on values in user manuals for Raven (Craig and the Raven Development Team, 2022) and UBC Watershed Model, as well as values used in BC Hydro models. |
| Snowpack cold content | CC_DECAY_COEFF | Calibration range was based on values suggested in UBC Watershed Model and values used in BC Hydro models. Constrained by calibration on SWE and discharge. Constraining features:
 • rate of snowmelt and associated runoff volume and timing after periods of cold weather |
| Snowpack patchiness | SNOW_PATCH_LIMIT | Calibration range was based on visual observations of snowpack patchiness and snowpack survey data. Constrained by calibration on SWE and discharge. Constraining features:
 • spatially averaged rate of snowmelt and associated runoff volume when snowpack is nearing complete melting |
| Soil layers and thickness | SoilProfiles | Calibration ranges based on soil mapping for the catchment. Constrained by calibration on discharge. Constraining features:
 • Runoff volume and timing, particularly in comparing sub-catchments with different soil distributions, and in comparing spring freshet runoff vs. rainfall runoff during snow-free periods
 • Simulated evapotranspiration and, thus, low flow volume are sensitive to soil depth |

| Soil texture and porosity | %SAND, %CLAY, %SILT, %ORGANIC, POROSITY | Values assigned based on soil mapping for the catchment. |
|---|---|---|
| Soils / runoff routing (infiltration/runoff partitioning, percolation, interflow, baseflow) | HBV_BETA, MAX_PERC_RATE, MAX_INTERFLOW_RATE, BASEFLOW_COEFF, BASEFLOW_N | Calibration ranges based values suggested in Raven user manual and input from Raven developers (J. Craig, personal communication, September 28, 2018). Constrained by calibration on discharge. Constraining features: <li>Runoff volume and timing (e.g., flashiness), particularly in comparing sub-catchments with different soil distributions, comparing sub-catchments vs. main catchment outlet, and comparing spring freshet runoff vs. rainfall runoff during snow-free periods</li><li>Shape of spring freshet hydrograph and rainfall driven event hydrographs</li><li>Rate of runoff recession at different points in time after peak flow</li><li>Low flow volume</li> |
| Potential evapotranspiration | PET_CORRECTION, PET_VEG_CORR | Constrained by calibration on SWE and discharge. Constraining features: <li>snowpack accumulation in forests vs. clearings</li><li>spring freshet and low flow runoff volume in 240 Creek Sub-catchment (mature forest dominated) vs. 241 Creek and Dennis Creek Sub-catchments (extensive forest cover disturbance), and in sub-catchments vs. main catchment outlet.</li> |
| Tree height | MAX_HT | Values assigned based on forest cover mapping. Model is insensitive to tree height. |
| Crown closure | FOREST_COV | Values assigned based on vegetation surveys and forest cover mapping. |

| | | |
|---|---|---|
| Leaf area index, forest shading | MAX_LAI, SVF_EXTINCTION | Calibrated for dominant forest types. Assigned for minor forest types based on calibrated results for dominant types. Model is generally insensitive to the minor forest types.

Calibration ranges based on vegetation surveys, forest cover mapping, and values in literature.

Constrained by calibration on SWE and discharge.

Constraining features:

• snowpack accumulation, snowmelt rate, and snowmelt timing in forests vs. clearings

• timing of hydrograph rising limb during spring freshet in 240 Creek Sub-catchment (mature forest dominated) vs. 241 Creek and Dennis Creek Sub-catchments (extensive forest cover disturbance), and in sub-catchments vs. main catchment outlet |
| Canopy interception | TFRAIN, TFSNOW, MAX_CAPACITY, MAX_SNOW_CAPACITY, MAX_INTERCEPT_RATE | Calibrated for dominant forest types. Assigned for minor forest types based on calibrated results for dominant types. Model is generally insensitive to the minor forest types.

Calibration ranges based on vegetation surveys, forest cover mapping, and values in literature.

Constrained by calibration on SWE and discharge.

Constraining features:

• snowpack accumulation in forests vs. clearings

• volume of spring freshet runoff and volume of runoff during rainfall driven events (e.g., rain-on-snow) in 240 Creek Sub-catchment (mature forest dominated) vs. 241 Creek and Dennis Creek Sub-catchments (extensive forest cover disturbance), and in sub-catchments vs. main catchment outlet |
| Leaf conductance | MAX_LEAF_COND | Values assigned based on those suggested in Raven user manual, and based on values used in BC Hydro models. |
| Terrain attributes | HILLSLOPE_LEN, DRAINAGE_DENS | Values assigned based on field observations, stream mapping, and values used in BC Hydro models.

Model is insensitive to these parameters. |

| Stream channel geometry and grade | SurveyPoints, Bedslope | Values assigned based on field observations and digitizing in Google Earth |
|---|---|---|
| Stream channel roughness | RoughnessZones | Initial values based on standard Manning's roughness coefficients, then allowed to vary in the calibration to match the travel time of flow between the sub-catchments and the main catchment outlet. |

The graphs used to present the results are clear and very useful. But it would be good if they had error bars to represent the range of results caused by equally good fitting model parameter sets.

- See relevant comments above and below.

I like it that the time series of the simulated and observed runoff are given for the individual years in the Supplementary material.

- Thank you for acknowledging the value of the time series. We would like to point out that many papers are published with much less detail related to the simulated and observed time series (e.g., fewer and/or much smaller plots are often provided). We provided this detail so the reader could clearly review the fit to the observed data, as the observed data were very important for constraining internal model processes. We believe that providing these time series for review adds credibility to the results.

The paper is long but overall, well written.

- Thank you for this comment. We agree that the paper is long. Because of its length and the large amount of results/data, we've had to put a large effort into refining the manuscript. This issue relates to some of our concern with adding content to the plots and text to address parameter uncertainty, as discussed above.

**Specific comments**

L14 and 862: Quantify this in a different way, e.g., in days or weeks. 2-9 times more is important if we talk about an advance of a week or several weeks due to disturbance but not if the advance is only 1 day.

- Good point. We propose revising the text to the following: "The combination of climate change and stand replacing landcover disturbance in the middle and high elevations is predicted to advance the timing of the peak flow two to nine times (depending on emission pathway) more than the advance generated by disturbance alone (7 days)."

L26: Maybe use a different word than values (hydrograph characteristics, hydrological signatures?)

- We are referring to values of concern to society, as changes in hydrology cascade into changes to watershed risk. We will work on clarifying the wording in the manuscript.

L76: Considering all the uncertainties in these assessments, the decimals are probably not warranted here.

- Good point. We'll remove the decimals.

L92: Give some info on the model here already. It would be good to know for the reader early on if you are using a physically based, spatially distributed model or some other model, if it was calibrated or not, etc.

- We'll add a couple points, in line with your suggestion.

L126-129: It is nice that you describe the vegetation here and give the codes that you will use for the vegetation codes throughout the text but it is hard for the reader to remember these codes, especially since there are also codes for the different scenarios. In other words, it would be a lot easier for the reader to understand the parts about the vegetation if you would just write out the names instead of using the codes.

- Okay. We'll go with your suggestion, provided the language does not become too cumbersome in the text.

L139: In addition to the mean annual runoff, also mention the mean annual precipitation, either averaged over the catchment or for at least one station. This is important information about the study site.

- Good suggestion. Thank you.

Section 2.2.1: It would be good to already mention how many HRUs there are in this section (now it is only mentioned on L243) and how many parameters there are per HRU. Now this section is short and a lack of knowledge on the model and its parameters early on in the paper, hampers the understanding of the other parts in section 2.2.

- Thanks for the suggestion. We think that it's best to discuss these details in the "spatial discretization" section, and would like to avoid redundancy. However, we will consider reordering Section 2.2 to have the meteorology section after the landcover and spatial discretization sections.

L195: How well is well? Is there a reference here or a result that you can add to the supp materials?

- We will provide a citation for Spittlehouse and Dymond (2022).

L267: How many parameters are there per HRU and in total? and how many of these were calibrated? Even after reading the paper, this is unclear to me. Please add this information clearly in the methods section. Ideally already in section 2.2.1.

- Each HRU had:
    - 13 soil / sub-basin runoff related parameters (8 calibrated)
    - 13 vegetation related parameters (6 calibrated for dominant forest types; all 13 assigned for minor forest types based on calibrated results for dominant types)
- There were also 30 parameters related to meteorology and energy balance (18 calibrated), and 12 parameters related to in-channel runoff routing (9 calibrated).
- See additional details provided in the table above.
- We will consider opportunities to provide this information concisely in the manuscript.

L268: What weighting did you use for the calibration? Equal for each of these objective functions?

- All components of the objective function shown in Table S3 (supplement) were assigned an equal weighting. This point will be added to footnote #1 for Table S3, and added to the text in the main body.

L281, 283 and ff: What exactly do you mean by constrained (or in L301 and 306 by informed)? Did you pick a parameter value a priori and not calibrate it or did you select a parameter range and calibrate within this range?

- Please see the table above for additional details on setting calibration ranges and calibrating on empirical data.
- The word "informed" was utilized to indicate that the data were used for setting calibration ranges (dominant forest types) and for assigning parameter values (minor forest types). We will revise the text to clarify these points.

L313: This wording is not clear. Did you use it to guess a specific value and then use this in the model? Did you calibrate within a certain range? A bit more information, or clearer wording would be useful.

- Field knowledge was used for adjusting calibration ranges. We will clarify this point.

L321: This is not clear - how did you get values for each specific channel? How different were these values?

- Google Earth was used for digitizing the width of the lower mainstem channel (10-15 m). Visual field observations were used for assigning the width of smaller channels in the upper reaches that were not clearly visible in satellite imagery (2-3 m wide). These points will be clarified in the text.

L331: How many parameters were optimized and how many were fixed based on field knowledge? Also did you use the same parameters for all the HRUs with the same vegetation or soil? Would it be possible to add a table with all parameter values and the range used for the optimization somewhere?

- See details above related to optimizing and assigning parameter values.
- All HRUs with the same vegetation type had the same parameter values for vegetation, and all HRUs with the same soil type had the same parameter values for soils. In this respect, two HRUs could have the same vegetation parameter values, but different soil parameter values, and vice versa.
- With respect to providing a table of parameter values and calibration ranges, we acknowledge the value in being able to review the parameter values. However, the primary author is a business owner in a competitive consulting environment, and catchment modelling of different climate and landcover scenarios forms an important component of his business. There is a large investment of intellectual property involved with parameterizing the model. There would be a considerable risk to his business competitiveness by publishing the model parameters.

L333, 389: How did you weigh these different objective functions in the calibration? All equal weight? Or did you optimize each function individually first? From L323-326, it appears that you did it sequentially? Or did you just use different time periods for each of these objective functions and calibrate everything at the same time using some weighted function? The current description doesn't make the calibration process very clear to me. Also, what is the reason for not using the NSE for the entire study period as well?

- As discussed above, all components of the objective function shown in Table S3 (supplement) were assigned an equal weighting, and all parameters were calibrated simultaneously. The language in L323 was intended to convey that different time periods were used for different data types (e.g., 1971-1981 for Penticton Creek discharge, 1984-1992 for discharge in the sub-catchments, and 1995-1997 and 2009-2014 for SWE). These varying periods of record were related to the availability of data in different time periods. These points will be clarified in the text.
- We should point out that the "clearcut" labels for UP9, UP11, and UP13 in Figures S2.5 and S3.5 should be labeled as "regen" (i.e., regenerating), consistent with Table S2. Also note that "leading species" in Table S2 should be changed to "vegetation type".

- Good question about NSE. It was not used for the entire study period because there is lower certainty in the quality of measured winter discharge related to potential ice build-up on the weir crest. Moreover, for constraining the model, the overall volume of runoff during the low flow period was more important than any short-term minor changes in flow, as the overall low flow volume relates to evapotranspiration and slow soil drainage / runoff processes. For these reasons, the decision was made to focus the low flow calibration on overall yield (i.e., absolute bias). These points will be clarified in the text.

L461: Already mention here if this is largely due to a change in precipitation or due to a change in evapotranspiration.

- Increasing evapotranspiration was an important cause of the decrease in net precipitation (P-E) in the 2050s, and increasing precipitation in the 2080s generated the partial recovery. This will be clarified in the text.

Figure 9: The shape of the curve changes as well. What is causing this? This requires some discussion.

- Good question. It is assumed that this point relates primarily to Figure 9b. The shape of the curve changes because there is a decrease in the peak flow for frequently occurring peak flows, and an increase for extreme peak flows. These changes are driven by the general decreases in snowpack accumulation (i.e., snowmelt runoff) and net precipitation, coupled with increasing extreme rainfall intensity. These points are made in the existing text; however, we will be sure to make a stronger connection to the shape of the curve.

Section 5.2.4: Make it clearer that this is the annual *average* discharge

- Good point. We will make that change.

L598: A lot of the quickflow probably consists of subsurface stormflow or even groundwater flow. The majority of quickflow is unlikely to be overland flow (surface runoff) for a forested catchment.

- Agreed. The point being made in this line relates to the forested condition generating an increase in event frequency for annual discharge, whereas it's the large burn that generates an increase in event frequency for peak flows.

L780: Groundwater would be a more likely source for the streamflow in the dry period than soil water (retention).

- Agreed. We will clarify this point in the text.

**Minor comments**

L11: Mention the name of the model or the type of model in the abstract.

- We'll add that point. Thank you.

L82-89: Move to the study site description.

- We believe this physiographic information provides helpful context upfront that landcover varies considerably in space and time.

L121: Explain that BEC is the biogeoclimatic ecosystem classification.

- Thank you.

L191: Lowest temperatures instead of coolest temperatures.

- We'll make that change. Thank you.

Figure 5: Maybe still add South and North to the axis labels for subpanel b?

- A few reviewers have requested clarification for this figure. We will rearrange the figure and adjust labelling to make it easier to interpret.

L431: These differences are very small. Highlight that first before giving the values!

- Good point. We'll make that change. Thank you.

L700: What do you mean by snowpack loads?

- We mean snowpack accumulation. We'll revise this for consistent wording.

L720-721: Explain better how this sentence fits here / what you mean by this? What is the link to the previous or next sentence?

- This point was provided to relate increasing rainstorm intensity to more rapid hillslope runoff, which would increase peak flows. We'll revise the text to clarify.

L797 values at risk: Do you mean the streamflow signatures / hydrograph characteristics? This could be worded more clearly.

- See related comments above.

---

## Author Comment (AC2)

**Author response to reviewer #2 comments for HESS manuscript "Modelling the effects of climate and landcover change on the hydrologic regime of a snowmelt-dominated montane catchment" [Paper #: hess-2024-361]**

Dear Reviewer,

We would like to thank you for taking the time once again to complete a thoughtful review of our manuscript submission. Addressing your comments will undoubtedly result in a stronger submission. Please find below a list of responses to your comments. We hope our responses satisfy the spirit and intent of your remarks.

Sincerely,

Russell Smith

**Reviewer #2 comments**

**General Comments**

Dear authors of the manuscript, thank you very much for revising the manuscript. The current version has undergone substantial enhancement in terms of content and presentation.

The manuscript is generally well-written, with figures and tables that are well-placed and clearly illustrate the results. The manuscript provides insights on annual and seasonal hydrological changes, as well as on the development of extreme summer low flows and peak flows under a range of climate scenarios and landcover conditions. The conclusions drawn are supported by the findings of the model. The in-depth analysis of potential future changes is informed by the CSIRO85 model. However, I still have some minor comments that should be addressed:

- Thank you for the very positive comments, and for acknowledging our large effort with the revisions.

**Minor comments**

L 12-14: what does "two to nine times more" mean in absolute terms? Absolute figures would provide more clarity here.

- Good point. We propose revising the text to the following: "The combination of climate change and stand replacing landcover disturbance in the middle and high elevations is predicted to advance the timing of the peak flow two to nine times (depending on emission pathway) more than the advance generated by disturbance alone (7 days)."

L 26-27: are the "values at risk" related to the society as mentioned in the first sentence of the abstract? This needs clarification, also in the conclusions section.

- Yes. We are referring to values of concern to society, as changes in hydrology cascade into changes to watershed risk. We will work on clarifying the wording in the manuscript.

L 267-268: which parameters have "substantial uncertainty and/or sensitivity"? Please name them. Although these parameters have been calibrated simultaneously, the issue of equifinality should be discussed (at least in section 6.4 on uncertainties).

- Parameters with the greatest uncertainty were conceptual model parameters, including those related to:
    - soils / runoff routing (HBV_BETA, MAX_PERC_RATE, MAX_INTERFLOW_RATE, BASEFLOW_COEFF, BASEFLOW_N; see Table A.3 in Craig and the Raven Development Team, 2022)
    - snowpack cold content and patchiness (CC_DECAY_COEFF, SNOW_PATCH_LIMIT; Table A.4)
    - potential evapotranspiration (PET_VEG_CORR; Table A.5)
- Parameters that were particularly sensitive include those related to:
    - short-wave radiation (CloudTempRanges, UBCCloudPenetration, MAX_LAI, SVF_EXTINCTION)
    - air temperature distribution (AdiabaticLapseRate, WetAdiabaticLapse)
    - precipitation distribution (PrecipitationLapseRate)
    - snowpack albedo (P0ALBMAX, P0ALBREC)
    - runoff routing on the soil surface and in the shallow soil (impacted flashiness during spring freshet; HBV_BETA, MAX_PERC_RATE, MAX_INTERFLOW_RATE)
    - runoff routing in deeper soil / groundwater (impacted rate of flow recession and maintenance of low flows; BASEFLOW_COEFF, BASEFLOW_N)

- - potential evapotranspiration in the middle and high elevation areas (PET_CORRECTION, PET_VEG_CORR)
  - canopy interception for snowfall (TFSNOW)
  - stream channel roughness (RoughnessZones)
- We believe the model was constrained well by the calibration process. This issue was discussed extensively in our response to Reviewer #1. We invite you to review those comments. We will address these concerns in greater detail within the discussion of uncertainties in the manuscript (Section 6.4).

L 303: I am still struggling with the meaning of these BEC variants. It would be nice if the BEC variants were explained in a table.

- We propose simplifying our description to the following: "Seven plots were established in mature forests ranging in elevation from 649 m (dry, ponderosa pine forest) to 1930 m (wet, sub-alpine forest) (locations provided in Figure 1)."

Table 5: would be helpful to split the numbers of net P into P and ET. The precipitation data given in other tables to not coincide with the aggregation used in Tab. 5

- We believe it is more important to present net precipitation in the main body of the manuscript, as it is more directly related to the availability of water for generating runoff. We're also hesitant to add another similar sized table to the main body for presenting precipitation and evapotranspiration, as the manuscript is already long; however, we propose adding a table to the supplement to present this additional information.

---

## Author Comment (AC3)

**Author response to reviewer #3 comments for HESS manuscript "Modelling the effects of climate and landcover change on the hydrologic regime of a snowmelt-dominated montane catchment" [Paper #: hess-2024-361]**

Dear Reviewer,

We would like to thank you for taking the time to complete a thorough review of our manuscript submission. You raised many important issues that will certainly result in a stronger manuscript. We disagree with some of your comments, but appreciate your thoughtfulness and value the constructive scientific discussion. Please find below a list of responses to your comments. We hope our responses satisfy the spirit and intent of your remarks.

Sincerely,

Russell Smith

**Reviewer #3 comments**

**Summary**

The authors describe a study to explore the effects of climate change and landcover change on the hydrology (SWE and various discharge variables) of a forested nival watershed in the interior of British Columbia. Results are derived via simulation using hydrologic modelling in combination with various forest disturbance scenarios and mid- and end-century climate projections. The research deals with an important and timely issue, particularly considering the dramatic increase in wildfire risk experienced in western North America. However, the work suffers from the strictly qualitative, predominantly graphical, approach to both analyse and interpret the results. This results in a work that generally lacks any real scientific impact. This is particularly puzzling considering that the experimental design employed (full factorial) and the available data lend themselves well to the application of common statistical techniques (ANOVA and regression) that could give quantitative results and analysis of individual and combined effects of climate and forest cover. This is a missed opportunity. I feel that with a more quantitative assessment of results, this work would make a worthy scientific contribution; hence, my recommendation is to accept but with major revisions.

- Thank you for acknowledging that the research deals with an important and timely issue. We agree!
- We respectfully disagree with your point that the work suffers from a strictly qualitative approach to analyzing and interpreting the results. We acknowledge that we followed a conventional/traditional approach in hydrology to presenting and interpreting the results; however, graphical plots are a means of presenting quantitative results, not qualitative results. Moreover, we discuss the patterns and effects quantitatively throughout the manuscript. If our work lacks any real scientific impact because of the approach to analyzing and presenting the results, then it should also be said that much of the existing hydrology literature also lacks any real scientific impact.
- Further to the points above, our work incorporated a substantive analysis in the frequency domain of peak flows, low flows, and mean annual discharge (i.e., annual yield). We're not aware of other hydrology literature that has examined all three of these hydrologic variables in this way. This approach led to valuable insights regarding contrasting effects on hydrology and associated watershed risks. We related these results to changes in climate, snowpack dynamics, and vegetation, which led to identifying mitigative influences. Moreover, three other peer reviewers have not had a concern with our overall analytical approach. We're confident that our work is scientifically meaningful.
- We acknowledge that the approach you've proposed for analyzing the model outputs is interesting and novel; however, not following your suggested approach certainly does not mean that our work lacks complexity or impact. Further, we have some uncertainties about the validity of your proposal:
  - It seems you are suggesting ANOVA and regression as a means of hypothesis testing. We question whether it would be statistically valid to do so, as outputs from a pre-calibrated deterministic model cannot be considered random samples from a population.
  - We can see some value in using ANOVA or regression in a descriptive sense to help understand the model; however, we question the utility of that approach when we have the luxury of simply modifying the climate forcings and/or landcover, and examining the predicted effects on hydrology directly. This is a luxury that does not exist with field studies, making ANOVA and/or regression necessary/appropriate in those cases; however, with field studies, it is also important to ensure a randomized sampling design.

**Major Issues**

**Experimental Design**

The experimental design, which is central to the study, is described in an ad hoc fashion. The authors are encouraged to re-organize the document so that the experimental design is clearer. And in fact, it should be noted that the authors employ what is formally known as a factorial experimental design, where multiple factors (climate state/change, disturbance level, elevation, etc.) are tested for their influence on the outcome of a response variable (peak flow, max SWE, melt-out date, etc.). The experiments could be described as follows:

- Phase 1:
  - a 13 x 2 design (26 categories) with 13 x climate states and 2 x disturbance levels
- Phase 2:
  - SWE: a 2 x 2 x 3 x 3 design (36 categories) with 2 x slope aspects (north and south), 2 x disturbance level (forest and clearing), 3 x elevations (low, middle and high) and 3 x climate states (current, 2050s and 2080s
  - Discharge: a 5 x 3 design (15 categories) with 5 x disturbance levels and 3 x climate states

Although the factorial design could have been exploited to formally explore main effects and interactions between the various factors, using such methods as ANOVA and regression, the authors opted instead for a qualitative and graphical approach and, I feel, missed an opportunity for a more impactful study.

- Thank you for your ideas on how to describe the experimental design. We will revise the text to describe the experimental design more clearly near the beginning of the manuscript.

I am also not convinced that the graphical results can be used to isolate each individual effect (disturbance and climate) as stated. For example, the 'disturbance effect' in Figure 10 shows the effect of each disturbance conditional upon various climate states. I.e. the disturbance effect is not independent of climate state.

- We agree that the disturbance effect is conditional upon various climate states. That was a key point of the analysis and interpretation, and the reason we presented the results the way we did – that is, to examine how disturbance effects vary with climate, and vice versa for climate effects.

As said prior, formal statistical approaches could be used to quantify the effect of each individual factor as well as the various combined effects. The authors could also take their existing results one step further. As each synthetic climate series is stationary, each 100-year series could be divided into non-overlapping decadal periods that would provide ten replicates per category, producing a full factorial design with replicates. Any experimental design text will explain how to more formally exploit this approach.

- You've proposed using formal statistical approaches to quantify various effects; however, it is important to acknowledge that the dependency of the disturbance effect on climate would need to be examined through incorporating an interaction effect (e.g., in the case of a regression model). To any extent the model residuals might violate the various requirements of regression (randomness, homoscedasticity, normality, independence, lack of correlation with the independent variable), the results and inferences of the regression modelling would be weakened, invalid, and/or biased/skewed. In contrast, directly examining the modelling outputs from various model perturbations leads to a more direct and unbiased interpretation of the various effects as represented by the catchment model.

**Phases:**

I am not clear on the requirement of the Phase 1 portion of the methodology. It seems that the main purpose was to select the climate experiment to be used for Phase 2. If so, the authors should be aware that they, perhaps inadvertently, chose the 'driest' scenario (see Table 1); only CSIRO85 projects slightly decreased precipitation in the 2050s and has the smallest precipitation increase in the 2080s. I would highly recommend that the authors redo the Phase 2 analysis with at least one additional climate experiment (perhaps the wettest one, e.g. MIROC85 or MPI45).

- This manuscript is a resubmission of a paper that was originally submitted to HESS in the fall of 2023 and later rejected (hess-2023-248). The original manuscript was submitted with the Phase 2 results only (i.e., only CSIRO85 had been modelled because of limitations in available funding). The original reviewers requested modelling of additional climate projections. We obtained additional funding and completed the modelling of four additional climate projections, and incorporated those results with CSIRO85 into Phase 1. The paper was already very long with many results to synthesize and package for the reader; thus, we did not think it would work well to complete a detailed (i.e., Phase 2) analysis of the results from all five climate models. We also did not have funding available for that amount of additional analysis. As it turned out, however, CSIRO85 generated climate effects on the spring freshet hydrograph that were

generally intermediary between the other climate projections in relation to peak flow timing and discharge, and hydrograph elongation; thus, we saw it as appropriate to present the CSIRO85 results for the detailed examination of disturbance and climate effects (Phase 2).

- We believe CSIRO85 was the most suitable selection for Phase 2, for the reasons discussed above. However, we acknowledge that CSIRO85 is relatively "dry" and, thus, might result in more severe climate effects on low flows compared to other climate projections, for instance. We are prepared to complete event frequency analyses for the outputs from the other four climate projections. We are hesitant to include a full presentation of these additional results in the main body of the manuscript because of the substantial length of the existing manuscript; however, the additional event frequency analyses could be provided in the supplement and discussed briefly in the uncertainty section (6.4).

**Minor Issues**

Line 24: Not sure 'three dimensional' is a suitable term for what you are describing. Perhaps 'multi-facetted'?

- Thank you for the suggestion. We will consider other language options.

Line 146: More specifically, Raven estimates cloud cover, etc. from T and P. The term 'accounts for' is a tad vague.

- Agreed. We will revise the wording to reflect that these components are estimated.

Section 2.2.2.1: If available, why wasn't data from the Penticton Airport station used directly as a model forcing?

- The model was originally set up to use both P1 and Penticton Airport data for model forcing; however, the Penticton Airport data generated nonstationarity in the model parameters over the ~45 year optimization record that could not be resolved. The nonstationarity was related to the temperature data. We inferred that the nonstationarity might have been related to urbanization, or possibly an instrumentation issue at the station. In either case, the nonstationarity did not occur when only P1 data were utilized.

Line 173: You are using incorrect nomenclature. What you are describing as "emissions pathways" are what should be called climate projections (i.e. a projection is produced from a combination of an emission scenario and a global climate model). Your design only includes two, not five, emissions pathways/scenarios, RCP4.5. and RCP8.5.

- Thank you for pointing this out. We will adjust our language accordingly.

Line 174: That bound "90% of projections" for what variable(s) over what region?

- They bound 90% of the air temperature and precipitation range of CMIP5 projections for the southern Okanagan of British Columbia (Supplementary material, Spittlehouse & Dymond, 2022). We will add this information and citation to the text.

Line 194: Confirm the adjustment is made seasonally, and not monthly.

- Seasonally (Spittlehouse & Dymond, 2022)

Line 199: What parameters are used, and then presumably adjusted, to describe the distribution of tasmin, tasmax, and precipitation.

- Daily values. Only Tmax and Tmin need adjustment.

Lines 249-254: The process being described seems to be better explained as a process if intersecting various layers as opposed to imprinting. In other words, discretizing HRUs is the process of intersecting five individual layers: sub-basins, BEC variants, disturbance history, vegetation type, and 2K x 2K grid (to limit HRU size to <= 4-km2).

- Agreed. We will revise the text accordingly. Thank you for the suggested language.

Lines 279-285: The use of the word "constrained" implies that these parameter values were adjusted during calibration. Do you really mean the values were estimated a- priori using observations?

- We're meaning that the parameters were calibrated on historical data and, thus, their values were constrained by the need to fit the simulated outputs to the historical observations. However, our language has generated some confusion among multiple reviewers. We will revise the text to be more specific with respect to assigning parameter values versus setting calibration ranges and calibrating on observed data.

Lines 301-306: This section implies the LAI and crown closure are closely related, however, in my mind they describe different vegetation characteristics. LAI is a vegetation property (i.e. leaf area density for individual plants) whereas crown closure is a stand property (the density of individual trees/plants). The authors seem to have conflated the two properties. Is LAI, then, a combined value of vegetation and tree density?

- Good question. Thank you for pointing this out. LAI and canopy closure do not specifically represent density, but are correlated to each other and to tree density. LAI can be both a property for individual plants or a stand property when averaged across space, just as tree height can express the height of an individual tree or the mean height of a stand (or a particular layer in the stand). For catchment modelling with Raven, the specific LAI related parameter is the maximum LAI of the vegetation type (i.e., stand level). We will revise the text to clarify that the specific parameter is maximum LAI, and that it relates to the vegetation type as a whole (i.e., stand level).

Line 329: Which years were used for calibration and validation?

- The years are shown in the supplement on the respective calibration plots (Figures S2.1 through S2.5) and validation plots (Figures S3.1 through S3.5). We will insert a reference to the supplement.

Line 332: How was the composite function constructed from the various indicators (I'm assuming it was the arithmetic mean). Were the individual metrics weighted? Show the mathematical description of the composite function.

- The composite function was calculated as the arithmetic mean of all components, and all components were assigned an equal weighting. These points will be added to footnote #1 for Table S3, and added to the text in the main body.

Line 350: Replace "emissions pathways" with "climate projections".

- We will make that change. Thank you.

Line 353-354: Would recommend instead saying "For each 100-year simulation the landcover state was static and the meteorology derives from a stationary climate state"

- Thank you for the suggested language. We will revise the text accordingly.

Line 356: May use "dynamic equilibrium" instead of "wet".

- We think that dynamic equilibrium would make sense to people with experience in hydrologic modelling, but might generate confusion for others. We propose revising the text to the following: "The first year of simulation was used as a warm-up (i.e., spin-up) to ensure soil wetness had reached a dynamic equilibrium (i.e., "wet") leading into the subsequent simulation years."

Section 2.5: There are two experiments being conducted to assess climate and landcover change on SWE, a point-scale experiment and a catchment-scale experiment. It's not clear, however, how the two are used, or which is experiment is being referred to in the results sections (5.1.2 and 5.1.3). For the point-scale experiment, where are the sample sites located? Which experiments supply the results for Table 5 and Figure 5? Are both experiments necessary?

- With respect to Section 5.1.3, both experiments (point scale and catchment scale) are a sensitivity analysis, as described in Section 2.5. As a sensitivity analysis, the point scale experiment does not represent specific locations in the watershed; however, it represents site conditions that are realistic for the watershed. Figure 5 is based on the point scale experiment. We will revise the text to clarify where point scale versus catchment scale data are being presented/discussed.
- Section 5.1.2 and Table 5 (net precipitation) are based on the same high elevation sites as those in the point-scale snowpack results (Figure 5). We will revise the text to clarify this point.
- We believe the point scale and catchment scale outputs are both valuable. The catchment scale plots show very clear changes for different climates, and help to convey a sense of broad, landscape scale changes. The point scale results facilitate a focus on specific contrasts, and lead to identifying specific mitigating influences.

Line 377: Day numbers are hard to interpret. It would be helpful to add the corresponding calendar dates, 172 = June 21 and 264 = Sep 21. Or just give the dates instead.

- The date associated with a specific day of year varies from year to year. We propose revising the text to the following: "lowest 30-day mean discharge between day of year 172 through 264 (approximately June 21 to Sept 21).

Line 385: Incorrect reference.

- This is the guidance provided in Section 1.4 of the Raven user's manual:
  - To cite Raven technical details for technical reports, this manual may be cited as:
    - Craig, J.R., and the Raven Development Team, Raven user's and developer's manual (Version 3.5), URL: https://raven.uwaterloo.ca/ (Accessed xxx, 2022).
- We will add a date of access to our reference.

Line 478: Both crown closure and LAI are used interchangeably to describe stand density. Terminology needs to be clarified and unified across the text.

- Good point. Both can be used to describe vegetation density at the stand scale, and the two variables are correlated. Generally, LAI is more impactful in the model than canopy closure, which is why LAI was referenced in line 478 and elsewhere. However, canopy closure was used to represent stand density in classifying mature stands for discretizing HRUs (lines 294-299), as LAI is not represented in the catchment-wide VRI forest cover mapping. This is the only section where canopy closure is referenced as a variable for stand density. We will revise the text to clarify these points, and to state that LAI is otherwise used to describe stand density in the analysis.

Line 503: This conclusion is not obvious from the results.

- In Figures 5b, 6e, and 6f, the disturbance effect is positive for almost all sites, but small or negligible for some sites at the low elevation. We will revise the text to reference these figures, and to clarify the exception for the low elevation.

Lines 515-516: This conclusion if also not obvious from the available figures.

- We agree. More specifically, the disturbance effect is variable at the middle elevation in the catchment scale analysis, and comparing to the associated canopy closure (Figure 2d) is not straightforward. Notwithstanding, with respect to mitigating influences, we believe it is helpful to communicate that the greatest disturbance effects in the middle elevations were associated with high density stands located in the MS zone, based on a detailed review of the output files.
- We propose revising the text to the following: "At the middle elevation, this disturbance effect was minimal for the vegetation types represented in the site scale analysis (i.e., moderate density P/F; Fig. 5e), and variable in the catchment scale analysis (Fig. 7d). The greatest disturbance effects in the middle elevations were associated with high density stands located in the MS zone (based on a detailed comparison of the data for Fig. 7d and Fig. 2d)."

Line 533: I don't think the figures support this conclusion that clearly.

- We agree. Comparing to the associated canopy closure (Figure 2d) is not straightforward. We propose revising the text to the following: "The climate effect was greatest for lower elevations, and for higher stand densities at middle and higher elevations (based on a detailed comparison of the data for Fig. 7b-c and Fig. 2d)".

**Tables**

Table 2: Do the LAI values in this table reflect the spatial variation in crown closure? Are LAI and crown closure correlated (see earlier comment)?

- There is a positive correlation between LAI and canopy closure. As described in the manuscript and discussed above, canopy closure was used for discretizing the HRUs. The relationship between LAI and canopy closure from the vegetation surveys was used for setting calibration ranges for LAI, then LAI was calibrated on SWE and discharge. The LAI values in Table 2 generally reflect the spatial variation in canopy closure, but not based on a specific equation/relation between the two. As discussed above, LAI is more impactful in the model than canopy closure, which is why LAI is presented in Table 2.

Table 4: Indicate in the table header weather the variable is a mean (Winter, Summer) or a maximum (Spring).

- Good point. It's the long-term mean of the total for winter and summer, and the long-term mean of the maximum for the specified duration for spring. We will revise the caption to clarify, and add info to the table header.

Table 5: Which experiment are these results from?

- Point scale, as discussed above.

**Figures**

Figure 5: Which experiment (point or catchment) do these results derive from? The categories (y-axis) on the panels take a while to interpret as they change between columns and are not explicitly identified. Also note that the 'Disturbance effect" and "Climate effect" are not strictly independent (disturbance effect is conditional on climate state and climate effect is conditional on disturbance state). Do the bars show the mean or the median (assuming catchment scale results) or single sites (point scale results).

- Please see response above regarding data independence.
- The existing caption states that these results are from the site scale snowpack sensitivity analysis, and that the data are the mean of annual maximum SWE (mm) and median timing of snowpack melt-out (day of year).
- Multiple reviewers have expressed concern regarding the interpretability of this figure and Figure 10. We will revise and add to the y-axis labelling, and make some rearrangements to the panel layout for clarity (e.g., swapping the disturbance effect and climate effect columns). We will also experiment with different bar fills and thicknesses to hopefully improve clarity.

Figure 10: The categories (y-axis) on the panels take a while to interpret as they change between columns and are not explicitly identified. Also note that the 'Disturbance effect" and "Climate effect" are not strictly independent (disturbance effect is conditional on climate state and climate effect is conditional on disturbance state).

- Please see response above for Figure 5.

Figure 12: Decimals missing on secondary y-axis labels. Recommend showing the summer period on the graph (i.e. as background shading).

- We assume you're referring to the missing decimals on the discharge y-axis. Thank you for noticing that error. We will correct it.
- Good idea about adding the shading. Thank you.